# Anchovy boom and bust linked to trophic shifts in larval diet

Rasmus Swalethorp [1,2,3] ✉, Michael R. Landry[1], Brice X. Semmens[1], Mark D. Ohman[1], Lihini Aluwihare[1], Dereka Chargualaf[2] & Andrew R. Thompson[2]

Although massive biomass fluctuations of coastal-pelagic fishes are an iconic example of the impacts of climate variability on marine ecosystems, the mechanisms governing these dynamics are often elusive. We construct a 45-year record of nitrogen stable isotopes measured in larvae of Northern Anchovy (*Engraulis mordax*) in the California Current Ecosystem to assess patterns in food chain length. Larval trophic efficiency associated with a shortened food chain increased larval survival and produced boom periods of high adult biomass. In contrast, when larval food chain length increased, and energy transfer efficiency decreased, the population crashed. We propose the Trophic Efficiency in Early Life (TEEL) hypothesis, which states that larval fishes must consume prey that confer sufficient energy for survival, to help explain natural boom-bust dynamics of coastal pelagic fishes. Our findings illustrate a potential for trophic indicators to generally inform larval survival and adult population dynamics of coastal-pelagic fishes.

Population fluctuations spanning orders of magnitude are a hallmark of coastal pelagic marine fishes[1,2]. The resulting population 'booms-and-busts' (respectively, periods of continuous high and low spawning stock biomass; SSB) can persist for decades[1,3] and have major implications for ecosystem health and human socio-economic well-being[4–7]. Despite the global importance of coastal pelagic fishes and over a century of research, the mechanisms driving these fluctuations have proven elusive[8–10]. Major hypotheses seeking to explain population volatility have variously postulated that bottom-up or top-down processes affecting the survival of eggs, early larvae, and juvenile and/or adult life stages ultimately drive SSB[8]. Hypotheses focusing on early larval stages, such as Johan Hjort's 1914 critical period hypothesis[11], are underpinned by the capacity of larvae to obtain adequate prey in order to prevent starvation, augment growth, and maintain body conditions that facilitate feeding and predator evasion. Although it is challenging to obtain in situ insight on the interactions between larval fish and prey at broad spatial and long temporal scales, increased sensitivity of geochemical measurements, coupled with long-term oceanographic sampling, now afford the tools necessary to link ecosystem processes (e.g., trophic ecology) with boom-and-bust population fluctuations.

There are over 140 species of anchovies worldwide. Some are targets of intense fishing and important food sources for humans[12]. For example, the Peruvian anchoveta (*Engraulis ringens*) is the single largest fishery in the world, and its abundance variability has profound implications for the South American fishery[13]. In the CCE, Northern Anchovy (*Engraulis mordax*, hereafter, anchovy) is currently not intensely targeted by fishing but is an important forage species that affects the reproduction and survival of myriad marine predators[14–16]. Despite being a primary focus of the California Cooperative Oceanic Fisheries Investigations (CalCOFI) program for seven decades, the mechanisms underlying the dynamic population fluctuations of anchovy remain unclear[14,15].

All anchovy life stages mainly occupy coastal areas of the CCE where adults feed on relatively large zooplankton species that tend to be more common during periods of nutrient-rich coastal upwelling[14,17,18] (Supplementary Fig. 1). Larval anchovy feed on smaller prey, mainly calanoid and cyclopoid copepod nauplii, calanoid copepodites and protists[19]. Historically, high anchovy abundances were associated with the negative phase of the Pacific Decadal Oscillation (PDO), when water temperature tended to be cool and upwelling

[1]Scripps Institution of Oceanography, University of California - San Diego, La Jolla, CA, USA. [2]NOAA Fisheries Service, Southwest Fisheries Science Center, La Jolla, CA, USA. [3]National Institute of Aquatic Resources (DTU Aqua), Technical University of Denmark, Kgs., Lyngby, Denmark. ✉e-mail: rswalethorp@ucsd.edu

high in the CCE[3,14]. However, this historical pattern broke down in the late twentieth and early twenty-first centuries, with recent SSB higher and lower during warm and cool years, respectively[15,20]. These observations underscore the lack of mechanistic understanding of anchovy population fluctuations, particularly in an increasingly warming ocean.

Changes in trophic ecology likely affect larval survival and, ultimately, recruitment (i.e., the transition from larval to juvenile life history stage)[10,21]. We used anchovy larvae from the CalCOFI program (Supplementary Fig. 1), collected between 1960 and 2005 coupled with high precision Compound Specific Isotopic Analysis of Amino Acids (CSIA-AA) to test the hypothesis that SSB (there are currently no reliable anchovy recruitment indices) is defined, among other factors, by trophic characteristics and efficiencies encountered during early larval life, resulting in boom-and-bust dynamics. During this 45-year window, anchovy experienced a prolonged period of high biomass (boom) between 1962–1987 and a prolonged period of low biomass (bust) between 1988–2003[22], with approximately a 3-fold difference in SSB between them (Fig. 1a). In 2005, anchovy SSB briefly rebounded[23] to past boom levels. We quantitatively demonstrate that population dynamics of anchovy follow changes in food chain length (FCL) indicative of changes in energy transfer efficiency from the base of the food web to young larvae (<3-week-old) in the California Current Ecosystem (CCE).

## Results and discussion

### Trophic changes from 1960 to 2005

To assess the trophic ecology of larvae across the study time window, we first built a non-continuous (larvae were not available in all years) 45-year time series of stable N isotopes ($\delta^{15}N_{Bulk}$, from 18–23 mm in standard length, SL larvae, <3 week-old). $\delta^{15}N_{Bulk}$ provides information on the isotopic signature of the assimilated diet (prey), integrating over days to weeks in young larvae. For this time series analysis and in the additional analyses referenced below, we used Bayesian autoregressive state space model (SSM) estimates of yearly values to account for both intermittent missing yearly observations (e.g., due to missing CalCOFI cruises) and high variability in measured values within years. To aid the visual assessment of correlations between the different time series, we also calculated pairwise cross-correlations; because these cross-correlations are done on SSM outputs where interpolated data from missing years are not independent, results should not be interpreted in a statistical significance context. The record revealed relatively stable $\delta^{15}N_{Bulk}$ during the 1962–1987 boom period, followed by a marked, transient decrease in $\delta^{15}N_{Bulk}$ around the start of the 1988–2003 bust period (Supplementary Fig. 2a). This sudden shift in $\delta^{15}N_{Bulk}$ suggests a major disruption in the N sources supporting larval anchovy at the onset of the major SSB transition.

Changes in $\delta^{15}N_{Bulk}$ integrate changes due both to source N at the base of the food chain and to the number of trophic enrichment steps to consumers (illustrated in Supplementary Fig. 3). To separate these effects, we analyzed individual larvae with CSIA[24] for three "trophic" amino acids (AAs; glutamic acid, alanine, proline) that fractionate N at known rates with each trophic transfer and two "source" AAs (phenylalanine, glycine, Supplementary Fig. 4) that remain largely unaltered, recording N at the base of the larval food chain[25–27]. The drop in $\delta^{15}N_{Bulk}$ at the start of the 1988-2003 bust period was explained by a −2.5 ‰ decline in source N in phenylalanine (Phe), the canonical source AA (Supplementary Fig. 2b). This suggests that either changes in the inorganic N sources or the community of primary producers (as N fractionation differs among species and depends on ambient nutrient concentrations[28,29]) precipitated this change. Using published trophic discrimination factors (TDFs) to interpret the trophic $\delta^{15}N_{AA}$ values[30], we generated two estimates of larval trophic level, which we define as food chain length (FCL). We use FCL to explore changes in energy transfer and efficiencies rather than an absolute metric of larval

trophic level. $FCL_{Glu-Phe}$ was based on Glutamic acid (Glu) and Phe, the two AAs most often used, and $FCL_{Trp-Scr}$ was based on all trophic and source AAs, generating a more robust estimate[31]. Coincident with the abrupt shift in $\delta^{15}N_{Bulk}$ at the onset of the 1988–2003 bust period, both $FCL_{Glu-Phe}$ and $FCL_{Trp-Scr}$ rose markedly (Fig. 1b, Supplementary Fig. 2c). Unlike the brief $\delta^{15}N_{Bulk}$ and $\delta^{15}N_{Phe}$ diversions, however, the changes in FCL indices persisted until 1992 before gradually normalizing towards the pre-shift 1960–1986 range. In 2005, when the anchovy population briefly returned to the boom period level, the $FCL_{Trp-Scr}$ also returned to just above the 1960–1986 boom period level.

The observed changes in FCL indicate that anchovy larvae and/or their prey experienced abrupt and prolonged changes in diet over the analyzed time period. This observation lends support to a hypothesis that larval trophic changes impact recruitment to the adult population, but the mechanism by which this occurs is unclear. A way to consider the implications of a changing FCL is in terms of energy transfer efficiency, from the base of the food web up to the larvae. To illustrate this, we converted $FCL_{Trp-Scr}$ into an energy transfer efficiency estimate (Fig. 2a), assuming an average gross growth efficiency (GGE) of 20% for the larvae and all heterotrophs lower in the food chain[32–34]. During the 1962-1986 boom period, energy transfer efficiency was on average 34% higher than during the subsequent bust period, suggesting that the amount of energy reaching the larval population from primary production during the population bust was substantially reduced (Fig. 2). In 2005, energy transfer efficiency rebounded to intermediate levels. While the temporal pattern in energy transfer efficiencies can be sensitive to our assumption of constant GGE, the changes are strongly correlated with anchovy SSB (cross-correlation $r^2 = 0.59$ at a 2-year lag).

### Energy transfer efficiency and survival

If the hypothesis that larval trophic efficiency mediates recruitment is correct, we would expect larval demography (growth and survival) to reflect the observed trophic shifts. To test this idea and explore at what point in larval development they are most sensitive to such shifts, we developed a time series of the ratio of large (10–20 mm SL) to small (5–10 mm SL) larvae in annual CalCOFI samples, assuming that this ratio encapsulates demographic shifts driven by trophically-mediated changes in larval growth and survival. During the 1988-2003 bust period, the size ratio rose to an order-of-magnitude higher level relative to the prior boom period (Fig. 1c). The ratio also declined immediately preceding the 2005 SSB rebound. Cross-correlation analysis indicates a strong, positive correlation between $FCL_{Trp-Scr}$ and the ratio between large and small larvae (0.84 at 0-year lag; Fig. 1d), and between energy transfer efficiency and larval size ratio (−0.82 at 0-year lag; Fig. 2b). Thus, when larval FCL is at its highest level in the time series (i.e., the greatest number of trophic steps), the relative abundance of small anchovy larvae is at its lowest level. Additionally, cross-correlation analysis indicates a strong negative correlation (−0.61 at 1-year lag) between the larval size ratio and SBB (Fig. 1e). The period from 1988 to 2003, therefore, marks fundamental changes in anchovy adult population size, larval size structure, larval FCL, and energy transfer efficiency.

There are three non-mutually exclusive explanations for the rise in the ratio of large to small larvae: (1) an increase in daily mortality of small (young) larvae (i.e., decreased survivorship of 5–10 mm larvae relative to 10-20 mm larvae), (2) an increase in the growth rate of large (old) larvae (i.e., increased growth rate of 10–20 mm larvae relative to 5–10 mm larvae), or (3) a reduction in daily mortality of large larvae. The latter two explanations are unlikely, as they suggest that larvae grow and survive well during time periods with both low larval trophic efficiency and poor recruitment to the adult population. Conversely, the first explanation fits well with the patterns we observed. It is also supported by the observation of Thayer et al.[23] that the ratio of anchovy larvae to eggs decreased from the late 1980s to the mid-2000s; that is, during the 1988–2003 bust period, relatively few larvae survived given the number of eggs observed. Taken together, the

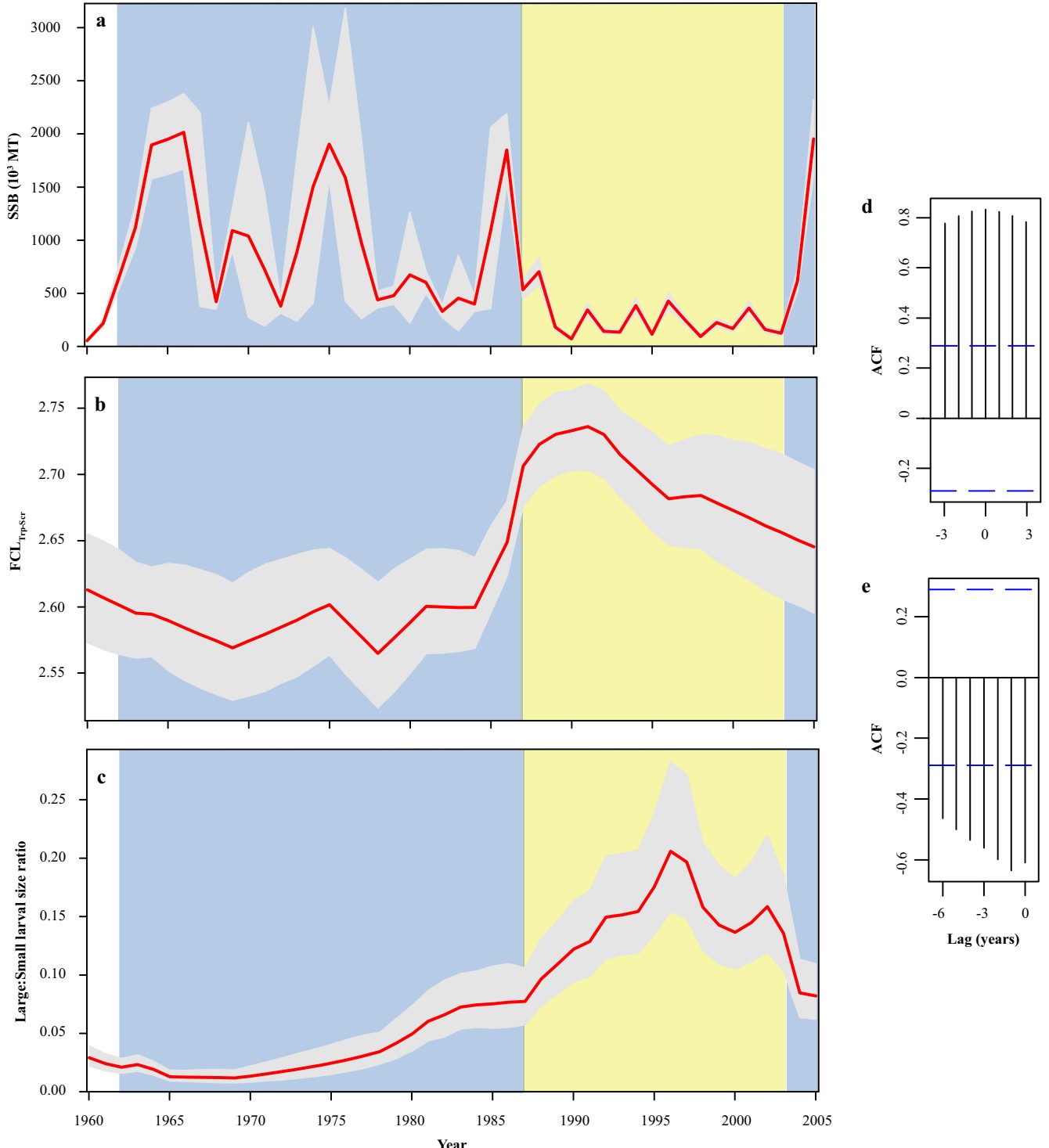

**Fig. 1 | 45-year time series of larval food chain length (FCL) and population dynamics.** Bayesian state space model estimated (**a**) spawning stock biomass (SSB) trends derived from annual estimates provided by Thayer et al.[23], with the 1962–1987 and 2005 boom periods shaded in blue, and the 1988–2003 bust period shaded in yellow, (**b**) $FCL_{Trp\cdot Scr}$ trends calculated from multiple trophic and source AAs for larval anchovy (*n* = 199), and (**c**) trends in the size ratio of all large (10–20 mm SL) to small (5–10 mm SL) anchovy larvae collected during spring from each station in the sampling grid (Supplementary Fig. 1). Red lines indicate maximum a posteriori yearly values, while error bands denote the 95% posterior credible intervals of yearly estimates for each metric. Maximum a posteriori cross-correlation analysis of (**d**) autocorrelation function (AFC) at year lag for $FCL_{Trp\cdot Scr}$ ~ Large:Small larval size ratio, and (**e**) Large:Small larval size ratio ~ SSB. Blue dashed bands indicate correlation threshold values (~ ±0.30) that would be met or exceeded 5% of the time by chance if the true lagged correlations were 0 (e.g., a white noise process with no correlation at any lag). Source data are provided as a Source Data file.

correlations among FCL, larval size ratio, and SSB suggest that early-life trophic dynamics mediate larvae survival to recruitment into the adult population, with survival during the first few days of life being most critical.

## Bottom-up or top-down effects

To gain further insight into what precipitated the anchovy boom-and-bust events, we next explored if environmental changes, ecosystem characteristics, and anchovy life history lend support to a hypothesis

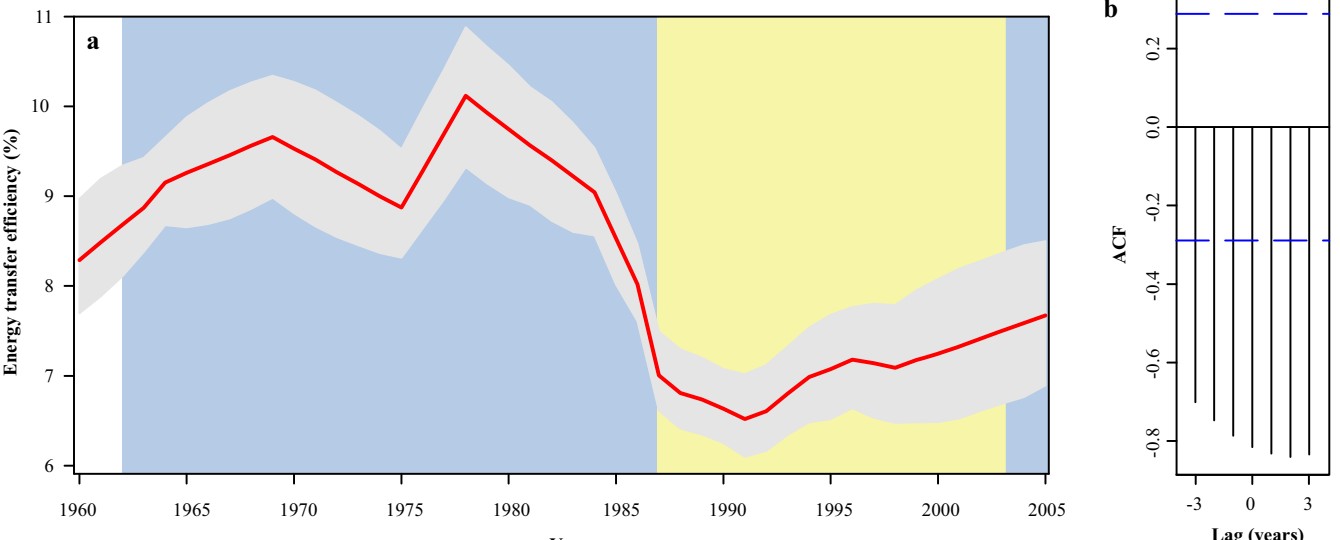

**Fig. 2 | 45-year time series of energy transfer efficiency.** Bayesian state space model (**a**) estimated annual energy transfer efficiency from the base of the food chain up to the larvae. The 1962–1987 and 2005 boom periods are shaded in blue, and the 1988–2003 bust period is shaded in yellow. Red lines indicate maximum a posteriori yearly values, while error bands denote the 95% posterior credible intervals of yearly estimates. Maximum a posteriori cross-correlation analysis of (**b**) autocorrelation function (AFC) at year lag for Energy transfer efficiency ~ Large:Small larval size ratio. Source data are provided as a Source Data file.

that low trophic efficiency encountered in early larval life could negatively impact recruitment. The CalCOFI program records several biotic and abiotic parameters at each station where larvae are collected, and others at select stations representative of the broader CalCOFI area. Of all environmental parameters and climatic indices explored (see Supplementary Table 1 for full list of variables examined), only zooplankton biovolume (of individuals <5 ml in size, routinely collected using Bongo nets with 505-μm mesh) related to FCL. Although this variable includes many organisms not preyed upon by anchovy larvae and misses smaller organisms that are, it coarsely serves as a temporal indicator of change in the broader prey community. At the start of the 1988–2003 bust period, zooplankton biovolume dropped considerably and remained lower than most of the prior three decades (Fig. 3a). Cross-correlation analysis showed a negative correlation between $FCL_{Trp-Scr}$ and zooplankton biovolume (−0.63 at 0-year lag). $FCL_{Trp-Scr}$ also correlates negatively with zooplankton biovolume for individual sampling stations (−0.29, Fig. 3b). Additionally, copepod and euphausiid biomass dropped to anomalously low levels at this time[35].

Calanoid copepodites are the main zooplankton prey of larger larval anchovy[19], and as with larvae of many other fish species, anchovy likely feed opportunistically on select species[10] among the diverse calanoid community of the CCE. We used pooled 505-μm mesh Bongo samples collected at select stations to explore whether calanoid community changes corresponded to changes in anchovy. Although this plankton net does not collect smaller nauplii stages nor does the analysis identify early copepodite stages also preyed on by the larvae it does provide a robust metric for the larger and mature copepod populations. Our data analysis revealed that a large change in the community structure occurred in 1989 (Supplementary Fig. 5). Several identifiable species decreased substantially in abundance at this time (Supplementary Fig. 5b), most prominently *Calanus pacificus* (Fig. 4a). Cross-correlation analysis reveals strong correlations between *C. pacificus* and $FCL_{Trp-Scr}$ (−0.75 at 0-year lag; Fig. 4b) and with the larval size ratio (−0.89 at 0-year lag). Our findings thus suggest that *C. pacificus* may be a critical food source for larval anchovy. This is not surprising given the importance of *Calanus* in the diet of other larval fishes[e.g.,10], due to their high lipid and energy contents[36,37]. In addition, Ala, the only trophic AA that enriches in $\delta^{15}N$ in heterotrophic

protists[38,39], is more enriched relative to Glu when FCL is longer (correlation coefficient 0.56), consistent with a lengthening of the heterotrophic protistan pathway in the larval food chain (Fig. 3c). Greater importance of heterotrophic protists is characteristic of lower production systems dominated by smaller phytoplankton, rather than productive upwelling systems typically inhabited by anchovy[40]. Food chains to the intermediate trophic levels of adaptive foragers like larval fish are often longer when productivity is low[41–44]. These observations suggest that trophic transfer efficiency to larval anchovy responds to changes in the zooplankton community as well as productivity at the food web base. The upward shift in anchovy FCL at the start of the 1988-2003 bust period was stable for at least six consecutive years, suggesting a prolonged period of suboptimal feeding conditions. Shorter-lived anchovy (maximal age is approximately four years) are capable of exploiting brief periods of high productivity opportunistically but are highly vulnerable to protracted periods of low production[14,15,40]. Thus, extended periods of low food and reduced energy transfer efficiency to larvae can detrimentally affect recruitment and spawning stock biomass. A highly efficient food chain has also been proposed as a key driver of anchovy production in the Humboldt Current upwelling system off Peru[45].

Shifts in zooplankton abundance can also impact juveniles and adults that inhabit and feed in the same environment and on similar zooplankton taxa as larvae[15,46,47], and adult fitness impacts egg production, quality, and size at hatch[48]. Maternal provisioning may be an important driver of larval Northern Anchovy survival[49], but it is unknown if it changed between 1960 and 2005 or if it influences larval FCL. Aside from the discussed bottom-up processes, top-down controls could exist on larval survival and SSB. Boom-and-bust dynamics of anchovies are known to have existed before significant human fisheries[2,50], but fishing could have contributed to population volatility in the time span of our study. Fishing pressure peaked in the late 1970s and early 1980s, when targeted removal of older more fecund individuals could have limited spawning of quality eggs[51,52]. Competition, cannibalism, intraguild predation or predation by higher trophic levels by larval, juvenile, and adult planktivorous fishes, carnivorous zooplankton, and other predators could also have impacted survival[53–55]. However, several known predators of anchovy, such as Pacific Mackerel (*Scomber japonicus*) and Pacific Hake (*Merluccius productus*), were

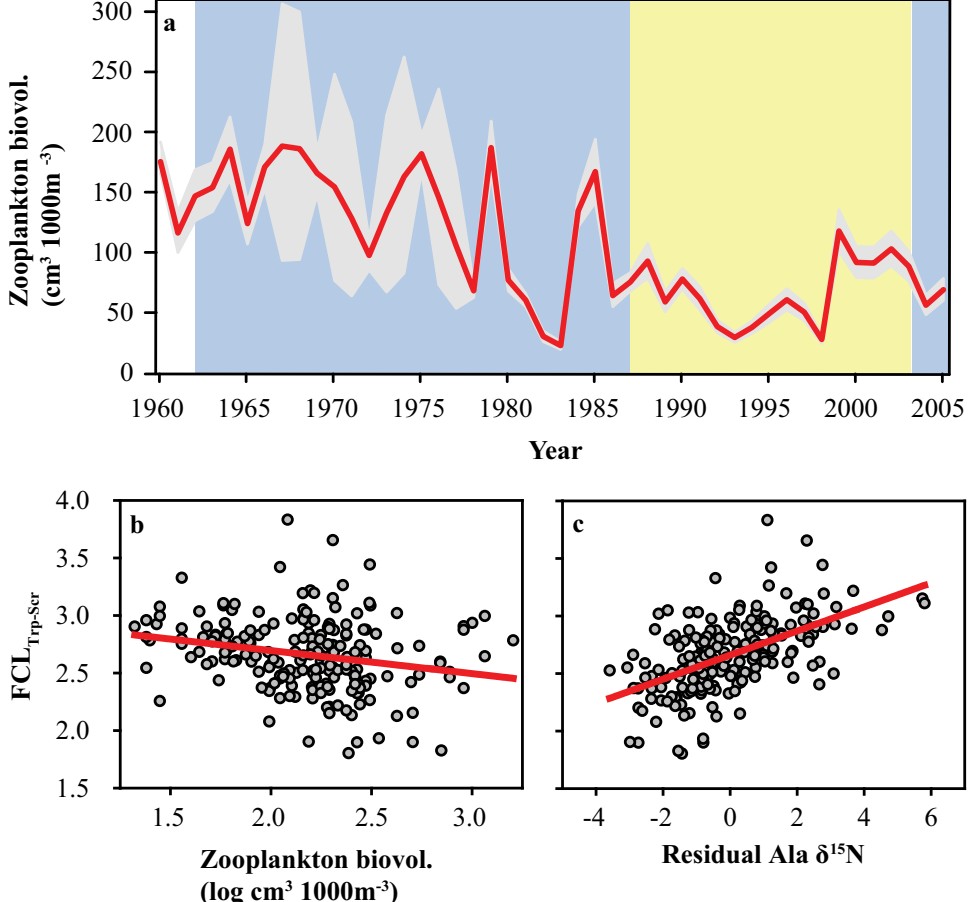

**Fig. 3 | Zooplankton time series and correlations. a** 45-year time series of the average zooplankton biovolume (organisms <5 ml) within the sampling grid (Supplementary Fig. 1) from spring CalCOFI cruises. The 1962-1987 and 2005 boom periods are shaded in blue, and the 1988–2003 bust period is shaded in yellow.

Correlation between $FCL_{Trp\cdot Scr}$ and (**b**) zooplankton biovolume ($n = 192$), and (**c**) residuals from the Ala $\delta^{15}$N - Glu $\delta^{15}$N relationship as a dimensionless indicator of heterotrophic protists contribution to the food chain ($n = 199$). Source data are provided as a Source Data file.

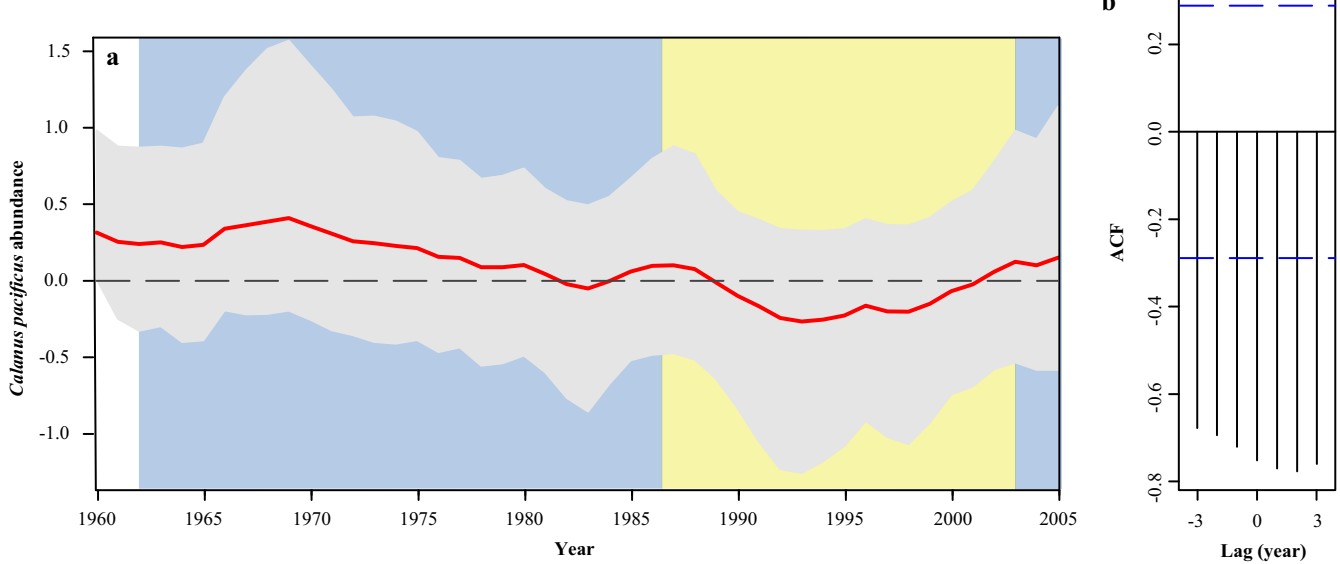

**Fig. 4 | *Calanus pacificus* abundance time series.** Bayesian state space model (**a**) of abundance z-scores from pooled samples at select stations in the CalCOFI area. The 1962–1987 and 2005 boom periods are shaded in blue, and the 1988-2003 bust period shaded in yellow. Red lines indicate maximum a posteriori yearly values,

while error bands denote the 95% posterior credible intervals of yearly estimates. Maximum a posteriori cross-correlation analysis of (**b**) autocorrelation function (AFC) at year lag for *C. pacificus* z-scores - $FCL_{Trp\cdot Scr}$. Source data are provided as a Source Data file.

abundant during the 1980s and their abundance dropped around the time that anchovy crashed[15,51,56,57]. Low zooplankton availability also could have led to increased food competition or exposure to intraguild predation[58,59]. It is conceivable that high competition/predation pressure could alter the behavior of the larvae affecting diet and FCL, but it appears unlikely that anchovy or other species could have sustained the high larval FCL and mortality that continued through the 1990s (Fig. 1c). A recent review by Sydemann et al.[15] found no consistent indications of competition or predation driving anchovy population dynamics throughout the studied period. Other CCE studies have noted that a major climate transition occurred in the late 1980s with physical-biological impacts[60–63] that percipitated zooplankton community changes and population changes across many species of larval, juvenile and adult pelagic and demersal fishes[60,64–67]. We thus cannot dismiss top-down effects on larval survival and SSB over the study period, but it is unclear how such mechanisms would affect FCL which is more directly linked to system productivity.

**Proposed ecological driver**

Based on these observations, we propose an explanation for the population fluctuations of coastal pelagic fishes: the Trophic Efficiency in Early Life (TEEL) hypothesis. TEEL hypothesizes that the high survival of young larvae depends on larvae feeding on prey that are part of a short food chain that maximizes energy transfer efficiency between the phytoplankton base and larvae. Figure 5 provides an illustration of the food chain structure characteristic of a population boom; high energy transfer due to a short and efficient food chain, low transfer through heterotrophic protists, supporting higher zooplankton and specifically *C. pacificus* abundance, and greater survival of young larval anchovies likely leading to greater SSB. Maximizing energy transfer efficiency would be increasingly

important during periods of suboptimal primary productivity in the larval habitat to support sufficient recruitment to maintain a high SSB. Our idea builds upon Hjort's[11] classic critical period hypothesis postulating that year class strength is contingent upon first-feeding larvae avoiding starvation by encountering prey. Later hypotheses explaining fish population dynamics based on larval prey such as match-mismatch[68], stable ocean[69], and optimal environmental window[70] focus on the oceanographic conditions that facilitate encounters between larvae and prey. TEEL builds upon these insights and adds another dimension describing the trophic characteristics of prey that reduce early larval mortality and enhance recruitment. Other factors, such as maternal provisioning, predation, and predator-prey interactions are likely co-contributors to larval survival and SSB dynamics[55,58], but to cite Hare[8] "The future of fisheries oceanography lies in the pursuit of multiple hypotheses". TEEL further provides another context to which indicators can be developed to inform stock assessments and management. Stock assessments generate estimates of adult SSB and recruitment, typically from surveys of adult biomass and size or age structure. Since adult structure cannot be applied to current-year recruitment estimates, this is often done by modeling spawning stock biomass against recruitment with Beverton-Holt or Riker curves, which are very poor predictors for coastal pelagic fishes (only 4% of global recruitment variability of small pelagics is explained by Riker or Beverton-Holt curves[71]). The FCL index could use existing larval data to improve near-term recruitment estimates and future population trends of anchovy and possibly other coastal pelagic species. To advance mechanistic understanding of TEEL and the forecasting benefits such ecological indicators may provide, we need to combine visual and molecular approaches to better identify the main prey species, and further explore how and where FCL is regulated in the food chain, its

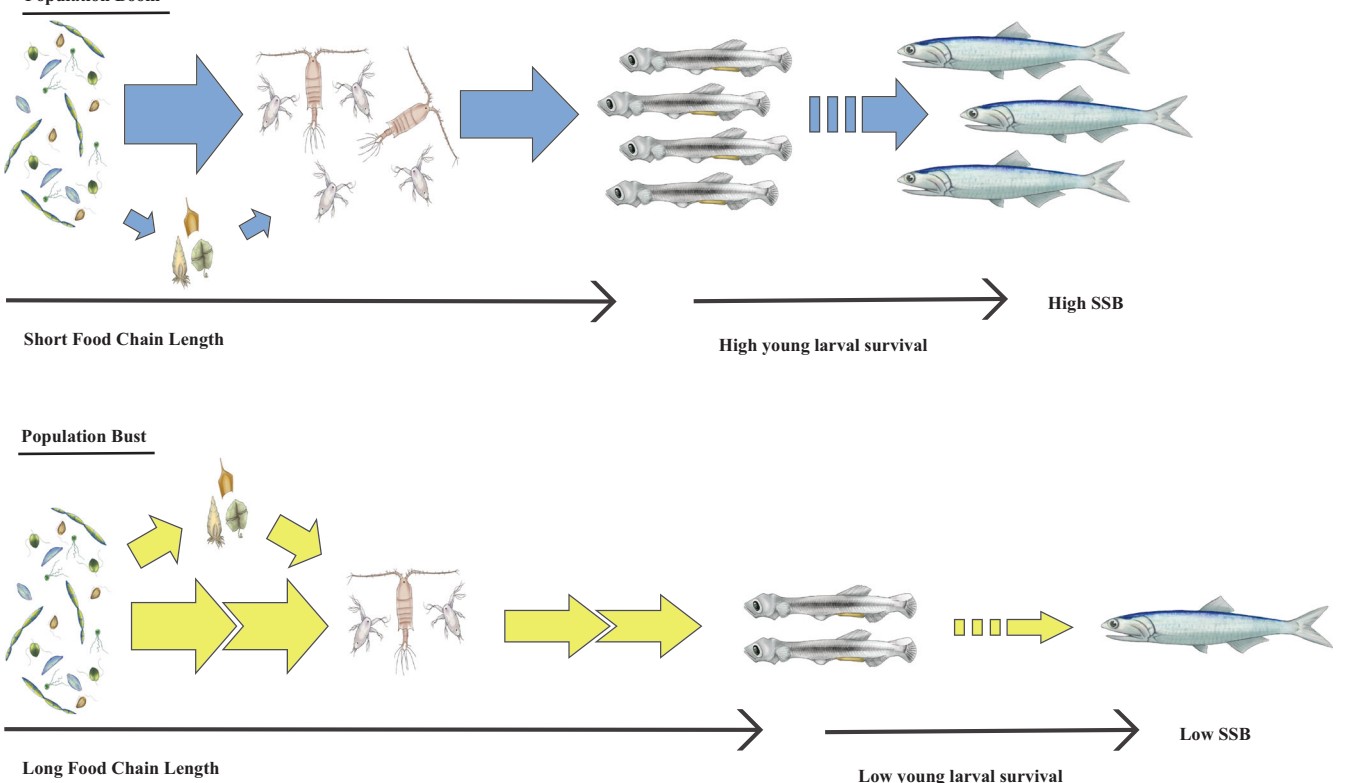

**Fig. 5 | Conceptual illustration of larval food chain length and anchovy boom and bust.** Arrow dimensions illustrate energy amount and distance traveled within the larval anchovy food chain, and the resulting recruitment. During population boom (blue), energy transfer is high due to a short and efficient food chain, with low transfer through heterotrophic protists supporting higher zooplankton biomass, and greater survival of young (small) larval anchovies leading to greater SSB. During the population bust (yellow) the situation is reversed. Organismal drawings provided by Dr. Claudia Traboni.

relationship to upwelling dynamics and its direct impacts on vital rates and survival. To achieve this, we need to collect samples that better resolve gradients in larval and plankton communities and interactions across space or time. Future research should also be aimed at investigating the applicability of TEEL to other fish stocks and how FCL measurements may be implemented for species of interest in near real-time.

## Methods

### Sample collection and preparation

Larval Northern Anchovy (*Engraulis mordax*; anchovy) was collected during spring CalCOFI cruises from 1960 to 2005 (www.calcofi.org). In this 45-year period, larvae were collected on 34 spring cruises. Collection was carried out by oblique tows to 210 m depth with Bongo nets of 0.71-m diameter, 0.505-mm mesh, and preserved in seawater with formaldehyde (1.3% final concentration) buffered with sodium tetraborate[72]. Larvae were later identified and sorted in the laboratory, standard length (SL) was measured under dissecting microscopes, and stored in individual 20-ml borosilicate glass vials at the sample archive at NOAA Fisheries Southwest Fisheries Science Center.

To avoid capturing a signal based upon maternal influence on stable isotopic signatures of the larval fish[73], we selected only the largest individuals from the archived samples for analysis. Analyzed larvae were in the standard length (SL) range of 18–23 mm. According to size ~ age curves, these larvae were 2–3 weeks old[74–79]. By selecting these size ranges, we ensured that the larvae had grown to ≥10 times their initial weight at yolk sack absorption, diluting the maternal isotopic signal, but still leaving sufficient larvae material for stable isotopic analysis (SIA) from a reasonable number of sampling years from the CalCOFI time series. These larvae were the survivors of the early larval stages that integrated dietary isotopic signatures over many days of feeding. Furthermore, it had been suggested that the late larval stage for anchovys is most important to survival[80], although this was not supported by our findings (see main article). In total, 207 anchovy larvae were analyzed for SIA over 20 Springs between 1960 and 2005 (Supplementary Table 2). For years of high larval catches, we focused on ≥3 core stations (defined as stations with the highest abundances of larvae in the desired size range), assuming these to have been collected from the most optimal nursery habitats and therefore most likely to contribute to recruitment. For years of low larval catches, we included larvae from all stations, assuming these to be the only ones that could contribute to recruitment. Within years and sampling stations, we took as wide a selection of larval sizes as possible within the defined size ranges.

All larvae analyzed were sorted from the archived samples and remeasured for SL. The heads, tail fin and internal organs including stomach and intestines were removed, and the remaining body transferred into individual borosilicate glass vials with polytetrafluoroethylene (PTFE) liner caps. The larvae were frozen at −80 °C, freeze-dried for 24 h, and stored in a desiccator until further processing. Samples were homogenized and subsampled (80–240 µg) by dry weight (DW) for bulk SIA. The remaining sample was kept for Compound Specific Isotopic Analysis of Amino Acids (CSIA-AA).

### Bulk stable isotopic analysis

Bulk SIA subsamples were transferred into tin capsules (Costech, 3.5 × 5 mm) and analyzed at the Scripps Institution of Oceanography Stable Isotope Facility (SIO). Samples were analyzed on a Costech ECS 4010 Elemental Analyzer coupled to a Thermo Finnigan Delta Plus XP Isotope Ratio Mass Spectrometer. Sample nitrogen ($^{15}N/^{14}N$) ratios were reported using the δ notation relative to atmospheric nitrogen ($N_2$). Measured $\delta^{15}N$ values were corrected for size effects and instrument drift using acetanilide standards (Baker AO68-03, Lot A15467). Long-term analytical precision of the instrument is ≤ 0.2 ‰.

### Compound specific isotopic analysis of amino acids

CSIA-AA was carried out using a novel, recently developed method where AAs are purified by High-Pressure Liquid Chromatography followed by offline Elemental Analysis–Isotope Ratio Mass Spectrometry (HPLC/EA-IRMS) of N isotopes[24,81,82]. This method is ideally suited for work on small samples and, unlike earlier isotopic analyses, has the necessary precision and accuracy to resolve the fine-scale $\delta^{15}N$ changes expected in planktivorous larval fishes[24]. We used three "trophic" AAs Glutamic acid (Glu), Alanine (Ala), and Proline (Pro) that enrich $\delta^{15}N$ at a known rate with each trophic transfer, and two "source" AAs Phenylalanine (Phe) and Glycine (Gly) that remain largely unaltered and reflect the inorganic source N at the base of the larval food chain[25–27].

For sample processing and AA purification, we followed the methodology described in Swalethorp et al.[24]. A minimum of 300 µg DW larval fish sample was hydrolyzed in 0.5 ml of 6 N HCl in the capped tubes for 24 h at 90 °C. The hydrolyzates were dried on a Labconco centrifugal evaporator under vacuum at 60 °C, re-dissolved in 0.5 ml 0.1 N HCl, and filtered through an IC Nillix – LG 0.2-µm hydrophilic PTFE filter to remove particulates. Samples were then re-dried before re-dissolving in 100 µl of 0.1% trifluoroacetic acid (TFA) in Milli-Q water, transferred to glass inserts in vials, and stored at −80 °C until AA purification.

AA purification was done with an Agilent 1200 series High-Performance Liquid Chromatography system equipped with a degasser (G1322A), quaternary pump (G1311A), and autosampler (G1367B). Samples were injected onto a reverse-phase semi-preparative scale column (Primesep A, 10 × 250 mm, 100 Å pore size, 5 µm particle size, SiELC Technologies Ltd.). Downstream, a 5:1 Realtek fixed flow splitter directed the flow to an analytical fraction collector (G1364C) and an Evaporative Light Scattering Detector (385-ELSD, G4261A), respectively. We used a 120-min ramp solvent program with 0.1% TFA in Milli-Q water (aqueous phase) and in HPLC grade acetonitrile (ACN, organic phase). The fraction collector was programmed to collect Glu, Ala, Pro, Phe, and Gly in 7 ml glass tubes at specified times based on elution times from previous runs. The quality of all collections was assessed by comparing chromatograms with set collection times, and only AAs where ≥99% of the peak areas fit within the collection windows were accepted. Injection volumes were determined from sample DW and the expected content of Phe, which was the least abundant AA of interest. Our aim was to collect ≥1 µg N equivalent of each AA. In order to assess the amount of AA collected, the ELSD was pre-calibrated by injecting different amounts of the liquid Pierce™ Amino Acid Standard H mix containing 17 AAs. Collected AA samples were dried in the centrifugal evaporator at 60 °C, dissolved in 40 µl of 0.1 N HCl, and transferred to tin capsules (Costech, 3.5 × 5 mm). The capsules were then dried overnight in a desiccator under vacuum. Pre-combusted borosilicate glassware, PTFE lines and HPLC-grade solvents were used in all process steps.

Isotopic analyses of AAs were done at the Stable Isotope Laboratory facility at the University of California, Santa Cruz (UCSC-SIL). Samples were analyzed on a Nano-EA-IRMS system designed for small sample sizes in the range of 0.8–20 µg N. The automated system is composed of a Carlo Erba CHNS-O EA1108 Elemental Analyzer connected to a Thermo Finnigan Delta Plus XP Isotope Ratio Mass Spectrometer via a Thermo Finnigan Gasbench II with a nitrogen trapping system similar to the configuration of Polissar et al.[83]. Measured $\delta^{15}N$ values were corrected for size effects and instrument drift using Indiana University acetanilide, USGS41 Glu and Phe standards and correction protocols (see https://es.ucsc.edu/~silab) based on procedures outlined by Fry et al.[84].

### Anchovy population data

Spawning stock biomass (SSB) estimates for the central stock of anchovy were acquired from Thayer et al.[23] for 1951–2015. Larval abundance and size data from CalCOFI were obtained from NOAA Southwest Fisheries Science Centers ERDDAP website

(https://coastwatch.pfeg.noaa.gov/erddap). We used these data to calculate a large (10–20 mm SL) to small (5–10 mm SL) size ratio. In calculating this ratio, we pooled all larvae from the central stock caught during spring between and including lines 76.7 to 93.3 in the CalCOFI grid (Supplementary Fig. 1).

## Environmental data

Zooplankton displacement volume (biovolume, $cm^3$ per $1000\,m^3$) from the same net tows in which the larval fish were caught was obtained from the CalCOFI website (https://calcofi.org/data/marine-ecosystem-data/zooplankton/). Zooplankton taxa abundances per $m^2$ from the spring cruises were downloaded from the ZooDB Zooplankton Database at the Scripps Institution of Oceanography Pelagic Invertebrate Collection (https://oceaninformatics.ucsd.edu/zoodb/, M. Ohman lab). These data were derived from analysis of pooled plankton samples collected on Southern California Bight lines out to and including station 70[35] (Supplementary Fig. 1).

The vast majority of anchovy larvae reside in the upper 30 m of the water column[85], therefore we averaged water temperature, salinity, $O_2$, chlorophyll a (Chl a) and $NO_3^-$ collected by CTD or bottles were averaged in the top 30 m. In addition, we obtained data on dynamic height, distance from shore, sea surface temperature (SST) from the CalCOFI database, and chlorophyll a (Chl a) from CTD and bottle data from CalCOFI cruises (https://calcofi.org/data/oceanographic-data/bottle-database/). In cases where CTD and bottle data were not available from the exact stations where the larvae were caught, we used data from the closest station within 18.5 km.

The Pacific Decadal Oscillation (PDO) Index was downloaded from the Joint Institute for the Study of the Atmosphere and Ocean (JIASO), University of Washington website (http://research.jisao.washington.edu/data_sets/pdo/#data). Multivariate ENSO Index (MEI) was obtained from the NOAA Physical Science Laboratory website (https://www.psl.noaa.gov/enso/mei.old/). The North Pacific Gyre Oscillation (NPGO) index[86] was downloaded from http://www.o3d.org/npgo/. The wind driven upwelling (Bakun index) and horizontal Ekman transport indices from latitude 33 °N, longitude 119 °W were obtained from NOAA Fisheries Environmental Research Division webpage (https://oceanview.pfeg.noaa.gov/products). A seasonal average for spring (March through May) was calculated for each index.

## Data analysis

The nitrogen isotopic compositions of the trophic (Trp) AAs Glu, Ala, and Pro, and the source (Scr) AAs Phe and Gly were measured consistently in 199 of the 207 anchovy larvae. $\delta^{15}N$ of these five AAs are not significantly affected by long-term formaldehyde preservation[24], and we, therefore, did not make any corrections to the values. Trophic positions, which we refer to as food chain length (FCL), were calculated using Glu and Phe, the canonical trophic and source AAs[87], and $\beta$ and TDF values from Bradley et al.[30] and Eq. 1:

$$FCL = \frac{\delta^{15}N_{Trp} - \delta^{15}N_{Scr} - \beta}{TDF_{AA}} + 1, \tag{1}$$

where $\delta^{15}N_{Trp}$ and $\delta^{15}N_{Scr}$ are the isotopic compositions of the selected trophic and source AAs, respectively. Nielsen et al.[31], among others, have advocated the use of multiple trophic and source AAs in calculating more robust estimates of FCL, which may be particularly important for formaldehyde-preserved samples[24]. Therefore, we also calculated a second FCL estimate using weighed means ($\bar{x}_W$) of all trophic and source AAs following Eq. 2:

$$\delta^{15}N_{\bar{x}_W} = \frac{\sum \frac{\delta^{15}N_x}{\sigma_x^2}}{\sum \frac{1}{\sigma_x^2}}, \tag{2}$$

where $\delta^{15}N_x$ is the value of a specific trophic or source AA and $\sigma_x$ is the procedural reproducibility error reported as standard deviation (SD) of that AA from four replicated analyses of Pierce AA standards[88]. These SD values were 0.24, 0.08, 0.25, 0.13, 0.19 for Glu, Ala, Pro, Phe, and Gly, respectively, obtained from Swalethorp et al.[24]. Weighed means were also calculated for the $\beta$ and TDF values from Bradley et al.[30]. Here, we report mean $\delta^{15}N$ and FCL values for each year of the time series ± the standard error (SE) of the mean. FCL was then converted into a trophic energy transfer efficiency between the phytoplankton base of the food chain and larval anchovy. Protozoans, metazoan zooplankton, and larval fish have gross growth efficiencies (GGE) of approximately 20–30%[33,89] while adult fishes have lower efficiencies of approximately 10%[32]. In calculating energy transfer efficiencies to the larvae, we assumed an average GGE of 20% with each trophic step, and that it remained constant. In our conversion of FCL to energy transfer efficiency we assumed that log10 to energy transfer had a linear relationship to FCL. However, GGE could vary in space and time with the composition of the communities involved, and energy transfer efficiencies would be sensitive to this.

Heterotrophic protists are potentially an important trophic link in the food chain of larval fishes[90]. Ala is the only AA that enriches in $\delta^{15}N$ in heterotrophic protists relative to their prey[38,39]. Therefore, we estimated an index of heterotrophic protist importance by fitting a linear regression to $\delta^{15}N_{Ala} \sim \delta^{15}N_{Glu}$ taking the residuals of $\delta^{15}N_{Ala}$. Since Glu, the most commonly used trophic AA, does not enrich in $\delta^{15}N$ in heterotrophic protists, we consider that deviations in $\delta^{15}N_{Ala}$ from $\delta^{15}N_{Glu}$ to be an indicator of the extent to which N is routed through heterotrophic protistan pathways in the larval food chain.

We tested the relationships among specific $\delta^{15}N$ values using Pearson product-moment correlation. Relationships among FCL and Phe $\delta^{15}N$, and environmental parameters and climatic indices listed in the previous Environmental Data section and Supplementary Table 1 were tested using Pearson product-moment correlation and linear mixed models (LMM) when relating parameters measured at the same sampling station at the same time. When using LMM we included sampling event (time and location) as a random effect. For parameters averaged seasonal or yearly we used cross-correlation analysis. Plotting was carried out in R or with Systat SigmaPlot v. 12.0 software.

We modeled the temporal trends in our time series response variables ($\delta^{15}N$, FCL, large:small larvae ratio, SSB, small fraction zooplankton displacement volume, *C. pacificus* abundance) using a Bayesian auto-regressive state-space approach[91], where measured values are assumed to represent samples from a true (unobserved) mean yearly value. We used this model fitting approach because it: (1) is explicitly auto-regressive (borrows information across years via hierarchical structure), (2) allows for multiple observations within years, and (3) accounts for years with missing data.

Our species-specific model formulation was as follows:

$$x_t = x_{t-1} + w_t, \text{ where } w_t \sim N(0, Q) \tag{3}$$

$$y_{i,t} = x_t + v_t, \text{ where } v_t \sim M(0, R). \tag{4}$$

Equation 3 presents the state equation, where $x_t$ represents a given mean response variable per year ($t$), and $w_t$ represents annual process error with variance $Q$. Equation 4 presents the observation equation, where $y_{i,t}$ represents the $i^{th}$ measurement of a response variable in year $t$. These observations are assumed to be normally distributed around the true mean response variable $x_t$ with observation variance $R$. All terms in the above equations are estimated within the modeling framework, with the exception of the $y_t$ observations.

All models were estimated using JAGS (Just Another Gibbs Sampler)[92] and the R statistical software environment. We assessed

model convergence by visually inspecting parameter trace plots and generating Gelman-Rubin potential scale reduction factors ($R_{hat}$)[93] using the CODA package in R[94]. In all model instances, we carried out sufficient iterations and thinning to ensure $R_{hat}$ values of less than 1.05 (adequate chain mixing and parameter convergence).

As a means of visualizing correlations between the modeled annual response variables, we carried out cross-correlation between the maximum a posteriori annual values at different yearly lags (using the CCF function in R). However, because some of our modeled response variable time series have missing time windows of data (notably, the stable isotope data due to lack of CalCOFI specimen collections), at least some of our mean annual response variables are interpolated using the Kalman filter within the SSM. As such, the lagged cross-correlation values generated from this analysis should not be interpreted through the lens of hypothesis testing and significance. Nonetheless, we feel these analyses provide a useful synthesis of both the apparent correlations between time series, and the time lags over which these correlations are strongest.

We created a map to visualize the distribution of anchovy larvae from the core CalCOFI stations by averaging samples collected between 1951 and 2016[56] (Supplementary Fig. 1). We used the package kriging to interpolate larval abundances between stations, then standardized (scale to zero mean and unit variance) the predicted values using the "decostand" function in vegan package[95] and plotted the standardized values with ggplot2[96].

We used chronological clustering analysis as implemented by the R package rioja[97]. The chronological cluster was computed using a bray-curtis distance matrix of log-transformed copepod data using vegan[95]. Deep breaks in the clusters were identified visually[98]. To visualize which taxa drove the chronological breaks, we created box plots showing the mean abundance of each copepod taxa in each cluster using ggplot2[96]. The box plots indicated that "small copepods" (less than 1.5 mm in length) and *Calanus pacificus* comprised the vast majority of the data. We thus explored relationships with anchovy variables and these two groups.

### Reporting summary

Further information on research design is available in the Nature Portfolio Reporting Summary linked to this article.

## Data availability

A summary table of all stable isotope data is available in Supplementary Table 3. Source data are provided with this paper both as raw data and state space model output data. CalCOFI hydrographic datasets used in this study can be found at https://calcofi.org/data/oceanographic-data/bottle-database/, larval anchovy population data at https://coastwatch.pfeg.noaa.gov/erddap/tabledap/, zooplankton community data at https://oceaninformatics.ucsd.edu/zoodb/, Pacific Decadal Oscillation Index from http://research.jisao.washington.edu/data_sets/pdo/#data, Multivariate ENSO Index from https://www.psl.noaa.gov/enso/mei.old/, North Pacific Gyre Oscillation index from http://www.o3d.org/npgo/, and wind driven upwelling index and horizontal Ekman transport indices from https://oceanview.pfeg.noaa.gov/products. Source data are provided with this paper.

## Code availability

The code for the State Space Model and cross-correlation analysis is available at https://github.com/BriceSemmens/ichthyoplankton_TS.

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

## Acknowledgements

We thank Magali Porrachia, Bruce Deck and Dyke Andreasen for their assistance in method development and sample analysis. We also like to thank William Watson, Julie Thayer and Peter Kuriyama for valuable input or comments on the manuscript, and Claudia Traboni for providing drawings for our conceptual illustration. This study was initiated with the Danish Council for Independent Research and the EU Marie Curie COFUND program Award DFF - 4090-00117 to R.S., and major portions were subsequently supported by a NOAA Fisheries and the Environment program (FATE) Award to A.R.T. and R.S., a NOAA RECOVER Award NA15OAR4320071 to M.R.L., a NSF-RAPID Award OCE-2053719 to B.X.S. and M.R.L., and NSF Awards OCE-1637632 and OCE-2224726 to the California Current Ecosystem Long-Term Ecological Research program (CCE-LTER).

## Author contributions

R.S., M.R.L. and A.R.T. conceived the project, and M.D.O. helped strategize sampling and methodology. D.C. prepared samples, and R.S.

performed stable isotope analysis and wrote the manuscript. L.A. and D.C. assisted in sample analysis. R.S., A.R.T., M.R.L., M.D.O., L.A. and B.X.S. contributed to data analysis and/or interpretations. All authors discussed the results and commented on the manuscript.

## Competing interests

The authors declare no competing interests.
