## [Peer Review File · Nature Communications]

Anchovy boom and bust linked to trophic shifts in larval dietEditorial Note: Parts of this Peer Review File have been redacted as indicated to remove third-party material where no permission to publish could be obtained.

REVIEWER COMMENTS

Reviewer #1 (Remarks to the Author):

Swalethorp et al review

The authors use stable isotope analysis to estimate food-chain length (FCL) of the diet of larval anchovy and investigate how FCL relates to recruitment success (actually stock biomass, SSB). The paper is super interesting and represent a major advancement in our understanding of the population dynamics of these fish. In particular, the link between SSB and larval diet is more satisfying than various correlations with basin-scale indices like PDO, and provides a more mechanistic connection, even if the exact functioning of the mechanism (beyond FCL) is unclear at present. Indices derived from this type of relationship are likely to be valuable for fisheries management and more reliable than relationships with PDO and other correlated predictors. This MS/paper should motivate additional work on anchovy as well as other species.

Some comments:

(1) The authors mention the adult's diet (line 58-59), but I could use a little more information on the larvae's diet. On what do the larvae feed? I'd suggest a couple of sentences in the main section to provide some foundation for the importance of the trophic transfer efficiency. There is some info in the methods on selecting older larvae to avoid maternal effects (which might be interesting in a separate paper), and prey size, but a couple of sentences max in the main section would improve the readability. Here (line 58-59) or a bit lower (eg line 67 or so) would be a good place to drop in a sentence or two on larval diet.

(2) The authors focus on older larvae to avoid maternal effects. This makes sense. However, maternal provisioning and egg size is important for other species, like rockfishes. Could similar effects of food-chain length influence egg production and provisioning and also contribute to the observed patterns? I'm not suggesting any additional analyses for this paper, but a few lines of discussion might be worthwhile.

(3) I know very little about isotope analyses, but nothing stands out as difficult to understand (from the text) or poorly done. However, the statistics are well done, especially the Bayesian state space model. The current model, however, appears to be density-independent, as $x(t) = x(t-1)$. Is there any reason to suspect that DD might be important?

(4) The SSM is great and, I think, the correct approach, but the authors might add a few sentences to the methods as to why, ecologically, we might expect an autoregressive effect on larval isotope signatures. Autoregressive effects make obvious sense with population abundance, but it isn't obvious why they would make sense for isotope signatures in 2wk old larvae.

(5) How might the assumption of constant rates of GGE (20%) affect the results? Could varying taxa in the food chain change that result, or is it good enough for a rough estimate. The authors note that fish GGE is lower (~ 10%) but I don't think larval anchovy are eating other fish, no? So the GGE of 20% is probably fine.

Line 22. "energy transfer efficiency to early larval survival, which in turn regulates the decadal-scale". Is 'regulate' the correct term? In an ecological setting it implies feedback through temporal density dependence. "Determines" might be a better word.

Line 48. "the food web to young larvae (< 3-week-old) over a 45-year". What are the larvae actually eating? Zooplankton or phytoplankton?

Line 104. "Changes in $\delta^{15}\text{N}_{\text{Bulk}}$ integrates". Should be 'integrate'.

Line 132. "unclear A" - I think you need a period between unclear and A.

Line 353. What is the size range of the non-analyzed larvae, just for comparison?

Line 540. I might add "interpolated via the Kalman filter within the SSM", or something like that for clarity. The interpolation is a little more complicated than just basic interpolation.

Extended Data Figure 4. If these are Bayesian results, shouldn't the intervals be "credible intervals" and not "confidence intervals"? (as in Fig. 1).

Reviewer #2 (Remarks to the Author):

This manuscript provides a very interesting analysis assessing the effects of changing food chain lengths as a potential driver of the boom-and-bust cycles of northern anchovy in the California Current Ecosystem over a 45-year period. Using a combination of bulk and compound specific isotopic analysis, measures of larval size ratios, zooplankton biomass, and oceanographic indices, the authors show that periods of high adult biomass are characterized by short food chains, high zooplankton biomass, high energy transfer efficiency, and lots of small relative to large larvae. The opposite patterns occur during a bust period of low adult biomass. The authors propose the Trophic Efficiency in Early Life (TEEL) hypothesis to explain how larval survival is impacted by ocean productivity, energy transfer, and food chain length in the pelagic ecosystem.

While the authors provide fairly strong support for the TEEL hypothesis, my primary concern is that there are only two periods represented in the data, one boom period and one bust period. The hypothesis would be much more convincing if there were multiple boom/bust periods and some replication of these shifting regimes to test whether food chain length, trophic efficiency, and large:small larval size ratios consistently change in the direction predicted as anchovy population dynamics shift. We are currently in another extended boom period, as the authors indicate, and CalCOFI data should be available more recently than 2005 to extend the time series into recent decades. Including more recent data from a separate boom period would provide much stronger inference, especially given the long gap in the stable isotope data set (1998-2005) at the end of the included time series. Why were more recent samples (post-2005) not included in the analysis? From Fig. 1, it appears that food chain length and large:small size ratios are decreasing at the end of the time series, following the predictions of TEEL, but this was not included formally in the analysis. How do the trends continue after 2005?

The change in bulk $\delta^{15}\text{N}$ and food chain length did not appear to be associated with a rapid change in the large:small larval size ratio. That ratio appeared to be gradually increasing prior to the bust years and while it increased more rapidly following the bust, it did so much

more gradually and later than the change in trophic indicators. Why?

Extended data Fig. 1 – legend could be more clear

Reviewer #3 (Remarks to the Author):

COMMENTS TO MS ID NCOMMS-22-32264-T

The submitted manuscript investigates the food chain length (FCL) and large : small size ratios for Northern anchovy (*E. mordax*) larvae caught in spring in the California Current system, analyzing a 45-year time series of isotopic analysis of amino acids. The authors present a novel hypothesis suggesting that high survival of young larvae depends on larval feeding on prey that are part of a short food chain length which maximizes energy transfer efficiency between trophic levels, leading to a higher small larval survival and therefore to a greater subsequent SSB.

Understanding key drivers affecting the recruitment of short life-cycle stocks, such as the present one, is an important subject, particularly within the context of an ecosystem approach to a sustainable and efficient fisheries management. To my opinion the study is, in general, well structured, presents relevant and useful information and uses an interesting approach, which might also be applied for other similar stocks worldwide. The use of language is also correct. However, there are some weaknesses that must be addressed before the manuscript can be considered suitable for publication.

The major issue regards on the new Trophic Efficiency in Early Life (TEEL) hypothesis. As it is presented in the current manuscript, I missed two main points. First, all the hypothesis is built in previous observations, and no actual sampling and/or data is incorporated, which would make the analysis more robust. In other words, one might expect some dedicated larvae sampling from, for instance, the last few years, for which a multi-proxy analysis could have been made, combining amino acid sampling with DNA metabarcoding and, perhaps also visual stomach content analysis under the microscope. That would have given a broader perspective of the TEEL, with some 'proved' and/or compared data from other approaches based on the same samples, and more detailed information that might help for the interpretation of historical isotope data. Otherwise, this is limited by the single method-based approach.

And this leads me towards the second caveat. While actual analysis might be considered as 'enough' to interpret the FCL and feeding efficiency, i.e., based on both CSIA and total prey biomass considering total zooplankton from samples caught within the CalCOFI program (the latter potentially driving, at least in part, the 'bottom-up' control mechanism), some relevant aspects that might affect the survival index of larvae as well as the recruitment are not considered. For instance, prey composition might be relevant, not only driven by larval size (i.e., prey size limited by larval gape size) but also by food availability (i.e., certain prey species might be more energetically interesting, such as eggs, Calanoid copepods.. in comparison with appendicularians, cylopoids or copepodites in general). I understand that incorporating new multi-proxy analyses might not be possible at this stage; but in such case, all commented points should be reflected better in the approach based on a more extended literature review (and therefore incorporated in the TEEL hypothesis). Larval survival might also be affected by the top-down control mechanism through potential predation by competitors (i.e., intraguild predation) or larger fish (predators from upper trophic levels). Further, other environmental factors, such as currents (mentioned in L.228-230) or other oceanographic conditions can also exert different pressures into larval survival or recruitment indices; these should be assessed more extensively (and if possible, providing some recent data & analysis combined with that suggested in the first point of this section).

Finally, the general approach should also be well linked to the sustainable fisheries management issue, not treated so extendedly throughout the manuscript. This might highlight the relevance of the study, pointing that further research might be applied within the context of the proposed TEEL hypothesis, also considering other pelagic fish stocks; or in contrast, what else should be checked to make it work also for other fish species.

Additional specific comments are included below ('L.' refers to 'lines').

Therefore, I kindly ask the authors to follow all these comments and clearly explain the main changes, or instead to give reasonable explanations to those points they decide to keep in its present form.

ABSTRACT

L.24-25: This last sentence seems state that both condition and recruitment processes can be explained with the approach, and that is not the case. Larval survival could be explained in part, and later processes are just hypothesized. I suggest changing to something like '...to inform on larval survival of coastal-pelagic fish populations'. Then I would also include one additional last sentence regarding potential applications of the approach, e.g. for spawning-stock biomass estimation, for fisheries management... Why is the study so relevant?

MAIN

L.36-39: In addition to these variables, predator and prey abundance could also be relevant for early life survival, through top-down and/or bottom-up control mechanisms. This aspect might fit well also within this section.

- L.44: I suggest changing to: ‘...that recruitment to adult populations is defined, among other factors, by trophic characteristics...’
- L.73-74: Why the authors did not include results until 2020? As mentioned in general comments, it would be nice to add some results & analyses of recent samples; dedicated sampling and detailed analysis on larvae (combining different approaches) would provide, with additional ‘proved’ evidence, an interesting information about ‘actual’ larval situation. That would be relevant to understand and interpret previous (historical) fluctuations. The authors might find the following multi-proxy approach as to be considered, at least for the methodological assessment: doi 10.1038/s41598-020-74602-y.
- L.76-87: I suggest including one sentence explaining what is behind $\delta^{15}\text{N}_{\text{Bulk}}$ analysis, i.e., providing some descriptive information about which type of information can be obtained might be helpful for potential readers not familiarized with this kind of approach. This would also help to follow the text in L.104-105.
- L.93-96: Please check & correct numbering of figures, either in the figure caption or changing the ordering of panels in the figure; $\text{FCL}_{\text{Top-Ser}}$ is figure (b), ‘trends in the size ratio...’ is figure (c); ‘spawning stock biomass...’ is figure (a)...
- L.105, Extended Data Fig. 2: These correlations should be treated with caution, considering such low coefficients ($R < 0.4$).
- L.111-112: ‘...changes in the inorganic N sources or the community of primary producers...’ Which primary producers? Please provide examples (and some references) here.
- L.114-115: ‘...we generated two estimates of larval food chain length (FCL).’ I suggest including an additional sentence explaining this term at first mention; clarify what FCL means and why is relevant for the purpose of the study. I understand all details are provided within the Methods section, but including such brief description in the main text might be helpful for the reader.
- L.118: Please check the numbering and labelling of figures; this seems to refer to Extended Data Fig. 4c instead of 3c...
- L.132: Please end the sentence (‘.’) after ‘...occurs in unclear’.
- L.133: Please add a comma: ‘...transfer efficiency₂ from the base...’.
- L.140-142: To state such a correlation between SSB, in addition to Figures 1 & 2, a statistical test is required.
- L.169-171: ‘...suggest that early life trophic dynamics mediate larvae survival to recruitment into the adult population...’ This must be supported by more analyses. Results presented show patterns for larvae but seems not enough to go further into conclusions for SSB. In addition to larval survival, many more drivers can also be relevant and affect SSB. The next section assesses food availability, which is fine, but there might be more: for instance, top-down control (intraguild predation [please consider including the following reference: doi 10.1007/s00227-015-2674-0], predation by fish from upper trophic levels...), or currents (i.e. predator-prey encounter rates, mentioned later in L.229-230; e.g., see also loophole dynamics in doi 10.1016/j.pocean.2007.04.011). All these issues might be appropriate for including in the text.
- L.185-186: Zooplankton availability/distribution analyses might be biased by 202 μm mesh size PRPOOS net sampling; this mesh size is adequate for relatively large prey, but not so efficient for smaller fractions (mentioned in L.187-188). Thus, more detailed information regarding zooplankton is required, such as the main taxonomic groups (either preferred by larvae or the most abundant); this would be possible incorporating multi-proxy approach suggested before, but instead, some more literature review might also be appropriate. Some zooplankton groups seem more energetically interesting than others, and prey species and size composition should have been assessed further. ZooDB Data base might also have some more detailed data that might be considered.
- L.190-191: This correlation coefficient is low, (< 0.3); the authors might explain or hypothesize about what other factors might be affecting the zooplankton biovolume.
- L.201: ‘...prolonged period of relatively poor feeding conditions.’ Which are those ‘poor feeding conditions’? Please provide more details and references (also related with previous comment for L.185-186). Some recent results (either presented as results from actual samples or based on recently published literature) are required here.
- L.205-206: ‘...zooplankton biovolume may also impact juveniles and adults...’ Please add more references here (i.e. also from other anchovy stocks, e.g. *E. encrasicolus*; doi 10.1093/icesjms/fss176; doi 10.3354/meps11375).
- L.206-207: This last sentence of the paragraph should be re-phrased; i.e., why late 1980s was a ‘tipping point’? Which were the most relevant changes in such ecosystem dynamics?
- L.221-222: ‘...low transfer through heterotrophic protists, supporting zooplankton biomass...’ What about zooplankton species and size composition? Also, this leads me to state one important issue: If food is in sufficient amount, an overlap in diet would not necessarily mean inter- or intra-specific feeding competition; this would also reduce the risk or larval

mortality due to intraguild predation or predation by other fish. This should be included somewhere in the manuscript (based on results from *E. encrasicolus*, already mentioned, doi 10.3354/meps11375).

- L.227-230: I already commented on this before, as general comments, but please highlight that additional factors than those assessed here may also affect SSB and recruitment (and provide references). Please include the intraguild predation and/or predation by other fish as a relevant component potentially driving larval survival. One sentence regarding the assessment of the approach, highlighting the strengths and the caveats of the methodology applied, might also fit well.
- L.232: I suggest closing the main text with a sentence about potential applications and relevance of the main findings (e.g., for ecosystem approach to fisheries management for coastal-pelagic species).

METHODS

- L.453-456: Were all those variables considered for the main analyses? Which are the variables significantly affecting zooplankton abundance (e.g., certain taxa or size ranges that would be more energetically interesting for larvae), larval abundance, potential predator abundance, etc.? A statistical analysis might be included, at least to justify which of them (i.e., not significant) are excluded from the main interpretation.
- L.507-520: Where temporal trends of oceanographic/environmental features (SST, salinity, Chl-a...) absent, or irrelevant for these response variables? I assume environmental variables were tested for relationships between FCL and Phe $\delta^{15}\text{N}$, but was their relationship with later SSB and recruitment data also tested?

Replies to reviewer comments

Reviewer #1 (Remarks to the Author):

Swalethorp et al review

The authors use stable isotope analysis to estimate food-chain length (FCL) of the diet of larval anchovy and investigate how FCL relates to recruitment success (actually stock biomass, SSB). The paper is super interesting and represent a major advancement in our understanding of the population dynamics of these fish. In particular, the link between SSB and larval diet is more satisfying than various correlations with basin-scale indices like PDO, and provides a more mechanistic connection, even if the exact functioning of the mechanism (beyond FCL) is unclear at present. Indices derived from this type of relationship are likely to be valuable for fisheries management and more reliable than relationships with PDO and other correlated predictors. This MS/paper should motivate additional work on anchovy as well as other species.

- *We thank the reviewer and indeed it is motivating additional work currently underway.*

Some comments:

(1) The authors mention the adult's diet (line 58-59), but I could use a little more information on the larvae's diet. On what do the larvae feed? I'd suggest a couple of sentences in the main section to provide some foundation for the importance of the trophic transfer efficiency. There is some info in the methods on selecting older larvae to avoid maternal effects (which might be interesting in a separate paper), and prey size, but a couple of sentences max in the main section would improve the readability. Here (line 58-59) or a bit lower (eg line 67 or so) would be a good place to drop in a sentence or two on larval diet.

- *We now added the following sentence: "Larval anchovy mainly feed on calanoid and cyclopoid copepod nauplii, small calanoid copepodites and protozoans."*

(2) The authors focus on older larvae to avoid maternal effects. This makes sense. However, maternal provisioning and egg size is important for other species, like rockfishes. Could similar effects of food-chain length influence egg production and provisioning and also contribute to the observed patterns? I'm not suggesting any additional analyses for this paper, but a few lines of discussion might be worthwhile.

- *It is possible that the FCL change observed in the larvae could occur in the adults as well, impacting their fitness and ability to produce quality eggs. In another paper currently in prep, we do find a strong relationship between anchovy otolith core size (as a proxy for size-at-hatch due to maternal investment) and larval condition, growth, and survival. As such, it may be that maternal investment accounts for much of the remaining SSB variability after larval trophic efficiency is accounted for. However, at present it is premature to draw such a conclusion, particularly since the samples needed to assess maternal investment for our study period do not exist.*

- *On page 9 we do write “Shifts in zooplankton biovolume may also impact both the juveniles and adults that inhabit and feed in the same environment as larvae.”. We have also added “, and adult fitness can impact egg production and quality (Riveiro et al, 2000)”. In response to reviewer 3 we have also added a new paragraph that mention maternal provisioning as an important driver of recruitment.*

(3) I know very little about isotope analyses, but nothing stands out as difficult to understand (from the text) or poorly done. However, the statistics are well done, especially the Bayesian state space model. The current model, however, appears to be density-independent, as $x(t) = x(t-1)$. Is there any reason to suspect that DD might be important?

- *We have thought about the potential role of DD on recruitment and chose not to include carrying capacity in the models because coastal pelagic species such as anchovy show little evidence of a consistent upper limit of abundance; rather, their booms and busts are notoriously stochastic, likely resulting from dramatic shifts in ecosystem states (as we have shown). Indeed, small pelagics are capable of producing strong recruitment classes when adult populations are either large or small. This point was wonderfully articulated by Reuben Lasker in several of his papers in the late 1970s and 1980s. Cury et al 2014 conducted a meta-analysis showing that Riker or Beverton-Holt models explain only 4% of the variability between SSB and recruitment:*

[FIGURE REDACTED]

We therefore chose not to include DD in our models.

(4) The SSM is great and, I think, the correct approach, but the authors might add a few sentences to the methods as to why, ecologically, we might expect an autoregressive effect on larval isotope signatures. Autoregressive effects make obvious sense with population abundance, but it isn't obvious why they would make sense for isotope signatures in 2wk old larvae.

- *Autoregressive models make sense in this case for two reasons: 1) the time series of isotope signatures show clear time-ordered patterns across years, suggesting each year should not be treated as independent of proximal yearly observations, and 2) for messy data with missing observations (years), as we have here, using a autoregressive model allows us to “borrow information” across years and ultimately more accurately represent the central tendencies of the time-varying process (Holmes et al 2020).*

Holmes, E.E., Scheuerell, M.D. and Ward, E.J., 2020. Applied time series analysis for fisheries and environmental data. NOAA Fisheries, Northwest Fisheries Science Center, Seattle, WA.

(5) How might the assumption of constant rates of GGE (20%) affect the results? Could varying taxa in the food chain change that result, or is it good enough for a rough estimate. The authors

note that fish GGE is lower (~ 10%) but I don't think larval anchovy are eating other fish, no? So the GGE of 20% is probably fine.

- *The results are sensitive to a change in GGE and indeed GGE can vary between different taxa. Here we are using a conservative GGE from broadly accepted averages for planktonic organisms and larval fish that we feel is robust enough to support our statement on changes in energy transfer efficiency. To further support our use of an average GGE of 20% we have now added an additional reference (Houde 1989) to a review with 20% as a conservative value across multiple studies on 10 species of larval fish.*

Line 22. “energy transfer efficiency to early larval survival, which in turn regulates the decadal-scale”. Is ‘regulate’ the correct term? In an ecological setting it implies feedback through temporal density dependence. “Determines” might be a better word.

- *Corrected. We now write “contribute to”.*

Line 48. “the food web to young larvae (< 3-week-old) over a 45-year”. What are the larvae actually eating? Zooplankton or phytoplankton?

- *They feed on a mixture of prey, but most of the ingested biomass is from zooplankton. In the following paragraph we have added this sentence: “Larval anchovy feed on smaller prey, mainly calanoid and cyclopoid copepod nauplii, calanoid copepodites and protozoans”.*

Line 104. “Changes in $\delta^{15}\text{N}_{\text{Bulk}}$ integrates”. Should be ‘integrate’.

- *Corrected.*

Line 132. “unclear A” - I think you need a period between unclear and A.

- *Corrected.*

Line 353. What is the size range of the non-analyzed larvae, just for comparison?

- *Non analyzed larvae were anywhere between 4 and 18 mm in length.*

Line 540. I might add “interpolated via the Kalman filter within the SSM”, or something like that for clarity. The interpolation is a little more complicated than just basic interpolation.

- *Corrected.*

Extended Data Figure 4. If these are Bayesian results, shouldn't the intervals be "credible intervals" and not "confidence intervals"? (as in Fig. 1).

- *Corrected.*

Reviewer #2 (Remarks to the Author):

This manuscript provides a very interesting analysis assessing the effects of changing food chain lengths as a potential driver of the boom-and-bust cycles of northern anchovy in the California Current Ecosystem over a 45-year period. Using a combination of bulk and compound specific isotopic analysis, measures of larval size ratios, zooplankton biomass, and oceanographic indices, the authors show that periods of high adult biomass are characterized by short food chains, high zooplankton biomass, high energy transfer efficiency, and lots of small relative to large larvae. The opposite patterns occur during a bust period of low adult biomass. The authors propose the Trophic Efficiency in Early Life (TEEL) hypothesis to explain how larval survival is impacted by ocean productivity, energy transfer, and food chain length in the pelagic ecosystem.

- *We thank the reviewer for the positive assessment of our study.*

While the authors provide fairly strong support for the TEEL hypothesis, my primary concern is that there are only two periods represented in the data, one boom period and one bust period. The hypothesis would be much more convincing if there were multiple boom/bust periods and some replication of these shifting regimes to test whether food chain length, trophic efficiency, and large:small larval size ratios consistently change in the direction predicted as anchovy population dynamics shift. We are currently in another extended boom period, as the authors indicate, and CalCOFI data should be available more recently than 2005 to extend the time series into recent decades. Including more recent data from a separate boom period would provide much stronger inference, especially given the long gap in the stable isotope data set (1998-2005) at the end of the included time series. Why were more recent samples (post-2005) not included in the analysis? From Fig. 1, it appears that food chain length and large:small size ratios are decreasing at the end of the time series, following the predictions of TEEL, but this was not included formally in the analysis. How do the trends continue after 2005?

- *The reviewer is correct that it would be very informative to include these more recent years and multiple boom-bust periods and it could add further support to TEEL. We do want to point out that around 2005 there was a brief resurgence in the population and FCL and size ratios had returned to past boom levels in 2005, but after that the population crashed again. So, our time series includes one long and one short boom period, and one bust period. In terms of why more recent years have not been included, the larval isotopic analysis for this paper was carried out between 2015-2017, and data not fully analyzed before 2020.*

- *In 2017 anchovy larvae had not yet been fully sorted from the CalCOFI samples, so it was still unclear how the larval anchovy population had responded in 2015-2016. The reason why no other years were included between 1998 and 2005, and between 2006 and 2014 was that, despite our extensive sampling efforts, there simply were not sufficient >18mm larvae available to analyze in those years from the CalCOFI samples. Anchovy abundances dropped to very low levels subsequent to 2005.*
- *Analyzing larvae from these more recent boom years (from 2015) are underway in connection with new projects with the aim of further advancing our mechanistic understanding of TEEL. Nonetheless, we feel that the current work provides sufficient evidence to put forward the proposal of TEEL, with the expectation that ongoing and future work will either refine the hypothesis or refute it. We have tried to temper our certainty regarding TEEL throughout the revised manuscript.*
- *A side note: We also have work ongoing on other species that lend support to TEEL. Work on the shortbelly rockfish *Sebastes jordani* showed that individuals with a lower FCL have significantly greater body fitness and significantly faster otolith growth histories (Kwan et al., manuscript.). Isotopic and modelling based work on Atlantic bluefin tuna aimed at understanding how the larvae survive in their highly oligotrophic open ocean nursery habitats has also shown that the larvae have very short FCLs (Swalethorp et al., in prep.) and that the larvae actively select prey that result in fast otolith growth histories and these short FCLs are lower than other cohabitating planktonic carnivores (Shiroza et al., 2021 and Stukel et al., 2021, Malca et al., 2022 in JPR). It could be that TEEL applies to other species with very different life histories and habitats than anchovy. Other studies have documented changes in trophic position in species of zooplankton across productivity gradients, and recently trophic efficiency has been speculated to be a key factor for anchovies in the Peruvian upwelling system (Massing et al, 2022).*

The change in bulk d15N and food chain length did not appear to be associated with a rapid change in the large:small larval size ratio. That ratio appeared to be gradually increasing prior to the bust years and while it increased more rapidly following the bust, it did so much more gradually and later than the change in trophic indicators. Why?

- *That is an excellent question. It is not clear what precipitated this more gradual increase in the size ratio prior to the sharp increase associated with the increase in FCL and population bust, or why the ratio continued its steep increase well into the bust period. The reviewer is correct that there does appear to be a slight lag in the extreme responses around the population bust. However, the strongest correlation between FCL and larval size ratio was found at 0 years lag.*

- *It may be that, in addition to FCL, size-at-hatch (maternal investment) also influences the large:small ratio. New research that we are working on (MS thesis by Robidas 2022, currently in prep for submission for peer-review) shows a very strong relationship between larval Northern anchovy otolith core size (which is a proxy for size-at-hatch as shown by Garrido et al. 2015) and larval survival, and body fitness. Between 1977 and 1984, SSB had dropped down to an intermediate level, and it is possible maternal investment was reduced during this time lowering the survivability of the larvae. The trophic conditions that precipitated the 1987 crash could also have progressively affected the adult's ability to produce larger larvae moving further into the bust period, and then progressively improved along with FCL moving towards the brief 2005 boom. However, we do not have samples of adult anchovy from these time periods, or preserved otoliths from larval anchovy (prior to 1997 CalCOFI samples were preserved only in formalin which degrades otoliths), so despite our extensive 45 years of samples and data this is not something that we will be able to test.*
- *Garrido, S., R. Ben-Hamadou, A. M. P. Santos, S. Ferreira, M. A. Teodósio, U. Cotano, X. Irigoien, M. A. Peck, E. Saiz, and P. Ré. 2015. Born small, die young: Intrinsic, size-selective mortality in marine larval fish. Scientific Reports 5:17065.*
- *Robidas, M. L. 2023. What drives larval condition for Northern Anchovy (Engraulis mordax)? Implications for coastal pelagic species recruitment fluctuations and fishery management practices. Masters Thesis. University of San Diego*

Extended data Fig. 1 – legend could be more clear

- *Corrected.*

Reviewer #3 (Remarks to the Author):

COMMENTS TO MS ID NCOMMS-22-32264-T

The submitted manuscript investigates the food chain length (FCL) and large : small size ratios for Northern anchovy (*E. mordax*) larvae caught in spring in the California Current system, analyzing a 45-year time series of isotopic analysis of amino acids. The authors present a novel hypothesis suggesting that high survival of young larvae depends on larval feeding on prey that are part of a short food chain length which maximizes energy transfer efficiency between trophic levels, leading to a higher small larval survival and therefore to a greater subsequent SSB. Understanding key drivers affecting the recruitment of short life-cycle stocks, such as the present one, is an important subject, particularly within the context of an ecosystem approach to a sustainable and efficient fisheries management. To my opinion the study is, in general, well structured, presents relevant and useful information and uses an interesting approach, which might also be applied for other similar stocks worldwide. The use of language is also correct.

- *We thank the reviewer for the positive assessment of our study.*

However, there are some weaknesses that must be addressed before the manuscript can be considered suitable for publication.

The major issue regards on the new Trophic Efficiency in Early Life (TEEL) hypothesis. As it is presented in the current manuscript, I missed two main points. First, all the hypothesis is built in previous observations, and no actual sampling and/or data is incorporated, which would make the analysis more robust. In other words, one might expect some dedicated larvae sampling from, for instance, the last few years, for which a multi-proxy analysis could have been made, combining amino acid sampling with DNA metabarcoding and, perhaps also visual stomach content analysis under the microscope. That would have given a broader perspective of the TEEL, with some ‘proved’ and/or compared data from other approaches based on the same samples, and more detailed information that might help for the interpretation of historical isotope data. Otherwise, this is limited by the single method-based approach.

- *Overall, reviewer 3 has done a great job outlining our decade-scale work plan and in the assessment that additional research is needed to further our mechanistic understanding of TEEL. As also outlined in our response to reviewer 2, we have a new project aimed at determining how and where FCL is regulated, if FCL impacts larval growth and how larval and young of the year FCL has behaved over recent years (2015 to now). In addition, we are 1) analyzing larval otoliths, 2) running eDNA analysis on CO1 and 18S to better characterize zooplankton communities that interact with larval anchovy, 3) running ROMS-informed Individual Based Models to try to better understand the oceanography that underpinned anchovy dynamics in the past 20 years, and 4) conducting these types of analyses for adults and other larval fishes such as rockfishes and sardine, and more. This work is all underway but in initial or mid stages.*
- *For now, we feel that the work presented here, using a uniquely comprehensive larval anchovy time series, has sufficient strength to support the proposal of a testable hypothesis. The exceptionally strong relationships that we demonstrate between larval FCL, mortality and SBB, and the linkages we can make to lower trophic level organisms collectively suggest that trophic efficiency is a key driver of larval survival and population fluctuations. We are excited to see future work address the hypothesis, regardless of the strength of support or refutation that manifests.*

And this leads me towards the second caveat. While actual analysis might be considered as ‘enough’ to interpret the FCL and feeding efficiency, i.e., based on both CSIA and total prey biomass considering total zooplankton from samples caught within the CalCOFI program (the latter potentially driving, at least in part, the ‘bottom-up’ control mechanism), some relevant aspects that might affect the survival index of larvae as well as the recruitment are not considered. For instance, prey composition might be relevant, not only driven by larval size (i.e., prey size limited by larval gape size) but also by food availability (i.e., certain prey species might

be more energetically interesting, such as eggs, Calanoid copepods.. in comparison with appendicularians, cylopoids or copepodites in general). I understand that incorporating new multi-proxy analyses might not be possible at this stage; but in such case, all commented points should be reflected better in the approach based on a more extended literature review (and therefore incorporated in the TEEL hypothesis).

- *We completely agree with the reviewer and have now added a copepod community analysis over the studied period to the manuscript. In the analysis we focus mainly on calanoid copepods in the Southern California Bight, as we know calanoids are a main prey of larval anchovy. We find that the community composition changed substantially right when anchovy was crashing in the late 1980s. Most larval fish are selective in their feeding, and anchovy likely only feed on a limited number of the many calanoid species that live in the CCE. We do not know exactly which calanoid species anchovy feed on, but a likely prey candidate is *Calanus pacificus*, as it is high in energy and *Calanus* is a preferred prey to many other larval fishes. *C. pacificus* is a dominant copepod species in the CCE, and, of the identifiable species in our analysis the most prominent in its changes. Its population changes match our anchovy indices and show an exceptionally strong correlation with FCL and the larval size ratio. We have now added the following text to the manuscript:*
- *“Copepod and euphausiid biomass were also dropped to anomalously low levels at this time (Lavaniegos & Ohman, 2007). Calanoid copepodites are the main group of zooplankton preyed on by larger larval anchovy (Arthur 1976). Further analysis of the calanoid copepodite community from pooled 505- μ m mesh net samples collected at select stations revealed a large change in community structure in 1989 (Supplementary Fig. 5). Of the several identifiable species with substantial abundance decreases at this time (Fig. 5), *Calanus pacificus* was the most prominent (Fig. 4a). Cross-correlation analysis revealed a strong correlation between *C. pacificus* and $FCL_{Trp-Scr}$ (-0.75 at 0-year lag; Fig. 4b), and with the larval size ratio (-0.89 at 0-year lag). Although we have no observational data of sufficient taxonomic resolution to support this, our findings suggest that *C. pacificus* may be a critical food source for larval anchovy. This is not surprising given the high preference often observed in other species of larval fishes for copepods of the *Calanus* genus (e.g., Robert et al., 2013), likely due to their high lipid and energy contents (Ohman 1997; Swalethorp et al., 2011).*
- *We also expanded the text on extensive climatic and population changes reported by other studies for many species fish and other organisms at higher and lower trophic levels in 1988/1989. These all provide further indication that the changes were driven at least in large part bottom-up processes.*

Larval survival might also be affected by the top-down control mechanism through potential predation by competitors (i.e., intraguild predation) or larger fish (predators from upper trophic levels). Further, other environmental factors, such as currents (mentioned in L.228-230) or other oceanographic conditions can also exert different pressures into larval survival or recruitment

indices; these should be assessed more extendedly (and if possible, providing some recent data & analysis combined with that suggested in the first point of this section).

- *We often harken back to the Conclusion of Reuben Lasker's last paper which states:*

"The answer to the question, "what limits clupeoids?," seems to be "almost everything." More realistically, the question should be phased "what limits clupeoids mostly?" Other questions follow this one: When, in the life cycle, does this occur? What are the interrelationships between limiting factors and between species? What can be learned from species life histories and fishery oceanography that will allow us to predict recruitment?"

- *We are not claiming here that we have solved the recruitment problem; rather, we have made a significant contribution to the field by identifying a novel, compelling, mechanistic hypothesis that has large implications for recruitment and population dynamics of an economically and ecologically important fish. Still, we acknowledge that other drivers are contributing to recruitment. We added a paragraph and text to other paragraphs discussing other possible control mechanisms. Please see additional explanation that we provide to the specific comments below. It's important to note that many of these controls have been explored in past studies and most recently reviewed in Sydeman et al, 2020, and so are not the focus of our study. The only mechanism that Sydeman et al, 2020 found to have a persistent (long term) relationship to SSB was upwelling, which is also the most likely mechanism of productivity and FCL.*

Finally, the general approach should also be well linked to the sustainable fisheries management issue, not treated so extendedly throughout the manuscript. This might highlight the relevance of the study, pointing that further research might be applied within the context of the proposed TEEL hypothesis, also considering other pelagic fish stocks; or in contrast, what else should be checked to make it work also for other fish species.

- *We added additional text to the last paragraph of the manuscript on these subjects.*

Additional specific comments are included below ('L.' refers to 'lines').

Therefore, I kindly ask the authors to follow all these comments and clearly explain the main changes, or instead to give reasonable explanations to those points they decide to keep in its present form.

ABSTRACT

L.24-25: This last sentence seems state that both condition and recruitment processes can be explained with the approach, and that is not the case. Larval survival could be explained in part, and later processes are just hypothesized. I suggest changing to something like '...to inform on larval survival of coastal-pelagic fish populations'. Then I would also include one additional last

sentence regarding potential applications of the approach, e.g. for spawning-stock biomass estimation, for fisheries management... Why is the study so relevant?

- *Thank you for this suggestion. However, the very short abstract limit required by the journal (ideally 150 words) prevents us from adding additional text. We have added more on this subject to the last paragraph of the paper.*

MAIN

L.36-39: In addition to these variables, predator and prey abundance could also be relevant for early life survival, through top-down and/or bottom-up control mechanisms. This aspect might fit well also within this section.

- *We have now added this briefly to the paragraph.*

L.44: I suggest changing to: ‘...that recruitment to adult populations is defined, among other factors, by trophic characteristics...’

- *Corrected.*

L.73-74: Why the authors did not include results until 2020? As mentioned in general comments, it would be nice to add some results & analyses of recent samples; dedicated sampling and detailed analysis on larvae (combining different approaches) would provide, with additional ‘proved’ evidence, an interesting information about ‘actual’ larval situation. That would be relevant to understand and interpret previous (historical) fluctuations. The authors might find the following multi-proxy approach as to be considered, at least for the methodological assessment: doi 10.1038/s41598-020-74602-y.

- *Thank you for suggesting Bachiller et al., 2020. This is an interesting paper and certainly an approach we will keep in mind in our ongoing projects.*
- *The logistical reasons for the time frame included in our study have been addressed in the prior responses above.*

L.76-87: I suggest including one sentence explaining what is behind $\delta^{15}\text{N}_{\text{Bulk}}$ analysis, i.e., providing some descriptive information about which type of information can be obtained might be helpful for potential readers not familiarized with this kind of approach. This would also help to follow the text in L.104-105.

- *We added the sentence “ $\delta^{15}\text{N}_{\text{Bulk}}$ provides information on the isotopic signature of the assimilated diet (prey) integrating over days to weeks in young larvae.”*

L.93-96: Please check & correct numbering of figures, either in the figure caption or changing the ordering of panels in the figure; FCLTrp-Scr is figure (b), ‘trends in the size ratio...’ is figure (c); ‘spawning stock biomass...’ is figure (a)...

- *Corrected.*

L.105, Extended Data Fig. 2: These correlations should be treated with caution, considering such low coefficients ($R < 0.4$).

- *Agreed. We merely included this figure to demonstrate the integrative nature of Bulk ^{15}N , i.e. that there is a significant relationship with different source and trophic AAs but that correlation coefficients are low with any individual AA. We have now clarified this in the figure text.*

L.111-112: ‘...changes in the inorganic N sources or the community of primary producers...’ Which primary producers? Please provide examples (and some references) here.

- *We do not have detailed species information of primary producers present during most of this time series, including when anchovy crashed in the late 1980s, or to what extent they fractionated N. However, it has been demonstrated in the literature that different taxa fractionate N to different degrees, and the degree a given taxa fractionates depends on the concentration of nutrients in its environment. We have now modified the text and included a couple reference to this point.*

L.114-115: ‘...we generated two estimates of larval food chain length (FCL).’ I suggest including an additional sentence explaining this term at first mention; clarify what FCL means and why is relevant for the purpose of the study. I understand all details are provided within the Methods section, but including such brief description in the main text might be helpful for the reader.

- *We added clarification as to why we choose this term.*

L.118: Please check the numbering and labelling of figures; this seems to refer to Extended Data Fig. 4c instead of 3c...

- *Corrected.*

L.132: Please end the sentence (‘.’) after ‘..occurs in unclear’.

- *Corrected.*

L.133: Please add a comma: ‘...transfer efficiency, from the base...’.

- *Corrected.*

L.140-142: To state such a correlation between SSB, in addition to Figures 1 & 2, a statistical test is required.

- *This has now been added. The correlation coefficient between energy transfer efficiency and SBB is 0.59 at 2-year lag.*

L.169-171: ‘...suggest that early life trophic dynamics mediate larvae survival to recruitment into the adult population...’ This must be supported by more analyses. Results presented show patterns for larvae but seems not enough to go further into conclusions for SSB. In addition to larval survival, many more drivers can also be relevant and affect SSB. The next section assesses food availability, which is fine, but there might be more: for instance, top-down control (intraguild predation [please consider including the following reference: doi 10.1007/s00227-015-2674-0], predation by fish from upper trophic levels...), or currents (i.e. predator-prey encounter rates, mentioned later in L.229-230; e.g., see also loophole dynamics in doi 10.1016/j.pocan.2007.04.011). All these issues might be appropriate for including in the text.

- *We thank the reviewer for these suggestions. The different factors brought up by the reviewer were not initially considered in this study as the focus was to develop new ecologically indices not previously considered, and that process led to the proposal of the trophic efficiency in early life hypothesis. We acknowledge, for instance, that we cannot rule out predation as a co-driver of anchovy boom-and-bust dynamics. We added a new paragraph that briefly lays out some of these factors that may have contributed to anchovy population dynamics, in addition to TEEL. We also added text pertaining to food availability and upwelling. Further, given the likely presence of other co-drivers, such as maternal provisioning, we acknowledge that we cannot conclusively say that FCL drives SSB; indeed the 0.39 unexplained variance suggests that there are more factors at play. Nonetheless, the 0.61 variance that is explained in the context of the long held view that early life history is important to recruitment does, we believe, justify the proposal that FCL has an impact on SSB. TEEL does not preclude the existence and impact of other controls mechanisms, nor do they preclude the existence of TEEL. We have modified the language to be a little less conclusive about SSB.*

L.185-186: Zooplankton availability/distribution analyses might be biased by 202 µm mesh size PRPOOS net sampling; this mesh size is adequate for relatively large prey, but not so efficient for smaller fractions (mentioned in L.187-188). Thus, more detailed information regarding zooplankton is required, such as the main taxonomic groups (either preferred by larvae or the most abundant); this would be possible incorporating multi-proxy approach suggested before, but instead, some more literature review might also be appropriate. Some zooplankton groups seem more energetically interesting than others, and prey species and size composition should have been assessed further. ZooDB Data base might also have some more detailed data that might be considered.

- *For clarification the zooplankton biovolume data presented comes from the same 505 μ m mesh sized Bongo net tows in which the anchovy larvae were collected, not the 202 μ m PRPOOS net tows. With the bongo data we have exact matches of zooplankton and larval information in space and time, while the PRPOOS net data does not overlap well in space with most of the larvae and has not been collected throughout most of our timeseries. From the Bongo data we have 2 sets of information available for the full timeseries, zooplankton biovolume from each individual sampling station, and pooled abundance information for CalCOFI sampling within the Southern California Bight of different zooplankton species. In the Environmental Data subsection of the Method we had wrongfully indicated that the zooplankton species data came from the PRPOOS net. This has now been corrected.*
- *We would like to thank the reviewer for the suggestion of taking a closer look at the zooplankton species and size composition information that is available on ZooDB. We had already included some data from ZooDB, which is outlined in the Methods – Environmental Data section and outlined in the Extended Data Table 3. Specifically, we tested if all or <1.5 mm copepods, crustaceans or all non-gelatinous organisms correlated with any of our trophic larval indices and found no relationships. However, we now include a more in-depth community analysis of the copepod community with special attention to the calanoids, the primary prey of larval anchovy, that shows that the community composition went through a major transition at the time that the SSB was crashing. Interestingly, *Calanus pacificus*, a likely prey to anchovy, correlates much stronger with FCL and the larval size ratio than bulk zooplankton biovolume. Please see our previous comment for details. Given that the community was sampled with a 505 μ m net, we do realize that we are missing many or the smaller younger development stages of species that comprises the larval diet. However, this is the best data available and an adequate indicator of prey community changes, and especially *C. pacificus*. We added additional text to the mentioned paragraph on these changes and cite another study that observed changes in copepod and euphausiid biomass concurrent with the anchovy population crash.*

L.190-191: This correlation coefficient is low, (<0.3); the authors might explain or hypothesize about what other factors might be affecting the zooplankton biovolume.

- *We present this relationship to indicate that something happened lower in the food chain, at the level of the zooplankton, that links with the observed changes in larval FCL. Included now is also the cross-correlation result between springtime zooplankton biovolume and FCL which has a higher correlation coefficient of -0.63. The reason for the weaker station-specific correlation (although significant) is likely that zooplankton biovolume is a very crude estimate of the fraction of the zooplankton community directly relevant as prey to larval anchovy as we state in the previous sentences “zooplankton biovolume (< 5 ml in individual size), which is routinely collected by CalCOFI using 505- μ m mesh sized Bongo nets, was one of only two parameters showing a relationship with FCL. Although this variable includes many organisms not preyed on by anchovy*

larvae and misses smaller organisms that are, it coarsely serves as an indicator of change in the broader prey community over time.” As stated above, the added copepod community analysis shows that the biovolume changes were associated with community changes, and the C. pacificus timeseries correlates even stronger with FCL than zooplankton biovolume.

- *Considering that the zooplankton, C. pacificus and anchovy populations were dropping at the same time and stayed at a much lower level for more than a decade, we argue this to be a bottom-up rather than top-down effect throughout this paragraph.*

L.201: ‘...prolonged period of relatively poor feeding conditions.’ Which are those ‘poor feeding conditions’? Please provide more details and references (also related with previous comment for L.185-186). Some recent results (either presented as results from actual samples or based on recently published literature) are required here.

- *Since we do not know the exact identity and availability of preferred prey species to the larvae when and where they larvae were around, we rephrased this statement to: ...” suggesting a prolonged period of changed feeding conditions.” We now also state earlier in this paragraph that this was a time when copepod biomass dropped to anomalously low levels (Lavaniegos & Ohman, 2007).*

L.205-206: ‘...zooplankton biovolume may also impact juveniles and adults...’ Please add more references here (i.e. also from other anchovy stocks, e.g. E. encrasicolus; doi 10.1093/icesjms/fss176; doi 10.3354/meps11375).

- *We added 2 references on adult Northern Anchovy diet and noted that adults feed on similar zooplankton taxa (although on average larger development stages) as the larvae.*

L.206-207: This last sentence of the paragraph should be re-phrased; i.e., why late 1980s was a ‘tipping point’? Which were the most relevant changes in such ecosystem dynamics?

- *We now rephrased the sentence to the following: “Other CCE studies have also noted that the late 1980s was a tipping point for several other larval, juvenile and adult species of pelagic and demersal fish.”*

L.221-222: ‘...low transfer through heterotrophic protists, supporting zooplankton biomass...’ What about zooplankton species and size composition? Also, this leads me to state one important issue: If food is in sufficient amount, an overlap in diet would not necessarily mean inter- or intra-specific feeding competition; this would also reduce the risk or larval mortality due to intraguild predation or predation by other fish. This should be included somewhere in the manuscript (based on results from E. encrasicolus, already mentioned, doi 10.3354/meps11375).

- *Zooplankton species and sizes are important in the context of what larval anchovy prefer to eat. However, we have very limited information on northern anchovy larval prey*

preference and on the taxonomic and size structure of the part of the zooplankton community within the larval prey size spectra. We are restricting our statement to only cover those variables where we have direct spatio-temporal overlap with larval anchovy and where our analysis revealed a relationship, where we can more definitively state that there is a connection.

- *We agree with the second part of the reviewers comment and have incorporated this into a new paragraph that appears just before this one.*

L.227-230: I already commented on this before, as general comments, but please highlight that additional factors than those assessed here may also affect SSB and recruitment (and provide references). Please include the intraguild predation and/or predation by other fish as a relevant component potentially driving larval survival. One sentence regarding the assessment of the approach, highlighting the strengths and the caveats of the methodology applied, might also fit well.

- *We included the following text at the end of the paragraph: “It is important to keep in mind that other factors, such as maternal provisioning and predation, are likely co-contributors to larval survival and SSB dynamics, but to cite Hare (2014) The future of fisheries oceanography lies in the pursuit of multiple hypotheses” and as we have demonstrated TEEL needs to take part in this pursuit. ... “To advance our understanding of TEEL and the nature of the information such ecological indicators can provide we need to further explore mechanistically how FCL is regulated and its direct impacts on vital rates and survival.”*

L.232: I suggest closing the main text with a sentence about potential applications and relevance of the main findings (e.g., for ecosystem approach to fisheries management for coastal-pelagic species).

- *We now included several sentences at the end of the paper discussing possible applications.*

METHODS

L.453-456: Were all those variables considered for the main analyses? Which are the variables significantly affecting zooplankton abundance (e.g., certain taxa or size ranges that would be more energetically interesting for larvae), larval abundance, potential predator abundance, etc.? A statistical analysis might be included, at least to justify which of them (i.e., not significant) are excluded from the main interpretation.

- *Yes, all these variables and more were considered and none were found to have strong significant relationships with any of the larval isotope-based indices except zooplankton biovolume and newly added *Calanus pacificus* abundances. Please see Supplementary Table 3 for the full list. This list also includes other taxa and size specific groups of*

*zooplankton, none of which were found to have a significant relationship to the larval indices. We explored further and found that of all the biological variables and indices tested only zooplankton biovolume showed any kind of relationship with environmental indices, a weak correlation with springtime PDO and MEI. Given the overall lack of relationships with environmental indices, we chose not to include this in the paper. Also, we do not have good data on total predator abundance and predation pressure on the zooplankton, so this was not explored. However, we now include a calanoid copepod community analysis that shows that the community and *Calanus pacificus* changed substantially around the time that the anchovy population was crashing.*

L.507-520: Where temporal trends of oceanographic/environmental features (SST, salinity, Chl-a...) absent, or irrelevant for these response variables? I assume environmental variables were tested for relationships between FCL and Phe $\delta^{15}\text{N}$, but was their relationship with later SSB and recruitment data also tested?

- *We consider temporal trends of oceanographic/environmental features to be outside the scope of our study because we are only able to resolve these seasonally on our quarterly cruises, and the larvae are only a couple of weeks old. That is., we consider the oceanographic/environmental features encountered at the same time the larvae were collected to be much more relevant to the larval response variables than features that occurred ~3 months prior. These variables were tested against FLC and Phe $\delta^{15}\text{N}$ and no relationships were found. Station-specific environmental variables (SST, salinity, chl a etc...) were not tested to yearly SSB estimates. Here we refer to the work of others and the recent review by Sydeman et al., 2020.*

REVIEWERS' COMMENTS

Reviewer #1 (Remarks to the Author):

The authors have addressed all my comments. This is a well done and very interesting paper. I advice publication as is.

Reviewer #2 (Remarks to the Author):

I have reviewed the revised submission by Swalethorp and colleagues examining the Trophic Efficiency in Early Life (TEEL) hypothesis to explain the dramatic boom and bust cycles of northern anchovy in the California Current Ecosystem. The authors have done an excellent job in revising the manuscript in response to my comments and those of the other reviewers and many portions of the manuscript have been greatly improved. The addition of the zooplankton community analysis and the focus on the abundance of *Calanus pacificus* as a potential driver of the boom-bust dynamics aligns well with their hypothesis. The only thing I might like to see addressed is why the bulk ^{15}N declined briefly at the bust transition but then rose back to normal levels. There is a lot of emphasis at the beginning of the manuscript on the how changes in ^{15}N signal the shift in anchovy population dynamics and a change in the food web, but the decline really only occurs for two years before returning to normal. Why? I suspect other readers may also seek at bit more explanation of that pattern. One small note, the order of the supplementary figures should probably be re-arranged as supplementary Fig. 1 is cited first (on line 55) and then supplementary Fig. 4 (on line 87), before referring to supplementary Figs. 2 (line 103) and 3 (line 106). Otherwise, the manuscript is excellent and I recommend it is accepted.

Reviewer #3 (Remarks to the Author):

COMMENTS TO MS ID NCOMMS-22-32264A.R1

The submitted manuscript investigates the food chain length (FCL) and large : small size ratios for Northern anchovy (*E. mordax*) larvae caught in spring in the California Current system, analyzing a 45-year time series of isotopic analysis of amino acids. The authors present a novel hypothesis suggesting that high survival of young larvae depends on larval feeding on prey that are part of a short food chain length which maximizes energy transfer efficiency between trophic levels, leading to a higher small larval survival and therefore to a greater subsequent SSB.

I am happy to see that most of my previous comments have been addressed thoroughly. I think the manuscript has been significantly improved.

Below I include some remaining issues to be addressed before the manuscript is suitable for publication.

MAIN

L.23-71: I encourage the authors to partially re-structure the paragraphs of this section, considering the following suggestion:

L.38-43: Move this part to the end of the subsection, and merge the text with that from L.63-71, as the latter is somehow redundant on summarizing the aim of the study and the main approach.

L.44-47: I suggest to remove these two sentences from the main text. Start the paragraph with '*In the CEE...*'.

L.200: The authors state, referring to Robert et al. (2014), that '*anchovy likely only feed on select species*'. However, to what extent the diet composition of anchovy is mainly driven by (active) feeding behavior (i.e., with active prey selection) or to opportunistic feeding on available prey is still open to discussion (many references report filter feeding as complementary feeding behavior on this species). Accordingly, I recommend changing the statement to something like '*...anchovy opportunistically feed on select species...*' (the authors can use the same citation as in the previous version, and even include some more if they find it appropriate).

L.207-209: According also to my previous comment, please change to '*...is not surprising given the importance of Calanus in the diet of other larval fishes^{e.g., 10}, due to their high lipid and...*'

L.220: Please add reference(s) after '*...is approximately four years*'.

L.273: Please check the figure number, did the authors want to refer it as Figure 5?

L.285: Please add some specific examples when mentioning 'predation'. Something like '*...provisioning and predation (e.g., intraguild predation^{e.g., 55}; [doi: 10.1007/s00227-011-1699-2], predator-prey interactions^{e.g., 58}; [doi: 10.3354/meps208229], etc.) are likely co-contributors to...*' might be appropriate here. Please note I include four references, two already used, and two additional ones with doi, as those could summarize different scenarios as examples. Please feel free to include more if appropriate.

L.289: Please remove 'typically' from the sentence (this term was already used in previous line).

L.296-300: According to my previous review, the authors still need to include more text regarding their methodological approach. I understand that a specific section might be too much, but at least some extended text in the Discussion regarding the major weaknesses of the methodology applied, limitations of the approach, and/or further potential improvements is required. Accordingly, in this part of the text the following issues must be treated (including appropriate references): zooplankton sampling bias (due to sampling tools' limitations to get the whole size range fractions of the zooplankton community that might be potential prey for anchovy larvae); potential application of a multi-proxy approach to test the hypothesis (i.e., combining visual prey identification from stomach contents of larvae, SIA and DNA-metabarcoding; use citation doi: 10.1038/s41598-020-74602-y); the need of dedicated sampling designs (how would the test of the hypothesis be improved by a new sampling design? Which aspects should be improved?); and how routinary sampling programs might contribute to improve the available data to apply similar approach to other regions and/or species?

REVIEWERS' COMMENTS

Reviewer #1 (Remarks to the Author):

The authors have addressed all my comments. This is a well done and very interesting paper. I advice publication as is.

- *We thank the reviewer for this positive assessment.*

Reviewer #2 (Remarks to the Author):

I have reviewed the revised submission by Swalethorp and colleagues examining the Trophic Efficiency in Early Life (TEEL) hypothesis to explain the dramatic boom and bust cycles of northern anchovy in the California Current Ecosystem. The authors have done an excellent job in revising the manuscript in response to my comments and those of the other reviewers and many portions of the manuscript have been greatly improved. The addition of the zooplankton community analysis and the focus on the abundance of *Calanus pacificus* as a potential driver of the boom-bust dynamics aligns well with their hypothesis. The only thing I might like to see addressed is why the bulk $\delta^{15}\text{N}$ declined briefly at the bust transition but then rose back to normal levels. There is a lot of emphasis at the beginning of the manuscript on the how changes in $\delta^{15}\text{N}$ signal the shift in anchovy population dynamics and a change in the food web, but the decline really only occurs for two years before returning to normal. Why? I suspect other readers may also seek at bit more explanation of that pattern.

- *We thank the reviewer for this positive assessment. Regarding the brief 2-year decline in Bulk $\delta^{15}\text{N}$ we did write in the manuscript: "The drop in $\delta^{15}\text{N}_{\text{Bulk}}$ at the start of the 1988-2003 bust period was explained by a -2.5 ‰ decline in source N in phenylalanine (Phe), the canonical source AA (Supplementary Fig. 4b). This suggests that either changes in the inorganic N sources or the community of primary producers (as N fractionation differs among species and depends on ambient nutrient concentrations^{28, 29}) precipitated this change."*
- *We do not know exactly what drove this 2-year pattern. There is no clear indication of a change in upwelled nitrate from wind driven upwelling indices or from the hydrographical data which could have affected the $\delta^{15}\text{N}$ baseline specifically during these two years. Also, the CalCOFI phytoplankton community data does not extend further back than 1996. So unfortunately, we cannot provide further clarity on this.*

One small note, the order of the supplementary figures should probably be re-arranged as supplementary Fig. 1 is cited first (on line 55) and then supplementary Fig. 4 (on line 87), before referring to supplementary Figs. 2 (line 103) and 3 (line 106). Otherwise, the manuscript is excellent and I recommend it is accepted.

- *Corrected.*

Reviewer #3 (Remarks to the Author):

[please see comments with formatted text in the attached document]

The submitted manuscript investigates the food chain length (FCL) and large : small size ratios for Northern anchovy (*E. mordax*) larvae caught in spring in the California Current system, analyzing a 45-year time series of isotopic analysis of amino acids. The authors present a novel hypothesis suggesting that high survival of young larvae depends on larval feeding on prey that are part of a short food chain length which maximizes energy transfer efficiency between trophic levels, leading to a higher small larval survival and therefore to a greater subsequent SSB.

I am happy to see that most of my previous comments have been addressed thoroughly. I think the manuscript has been significantly improved.

- *We thank the reviewer for this positive assessment.*

Below I include some remaining issues to be addressed before the manuscript is suitable for publication.

MAIN

L.23-71: I encourage the authors to partially re-structure the paragraphs of this section, considering the following suggestion:

L.38-43: Move this part to the end of the subsection, and merge the text with that from L.63-71, as the latter is somehow redundant on summarizing the aim of the study and the main approach.

- *This text has now been incorporated into the aim and approach at the end of the main section.*

L.44-47: I suggest to remove these two sentences from the main text. Start the paragraph with 'In the CEE...':

- *This text provides important context to the general importance of small pelagic forage fishes to global fishery. We prefer to keep it.*

L.200: The authors state, referring to Robert et al. (2014), that 'anchovy likely only feed on select species'. However, to what extent the diet composition of anchovy is mainly driven by (active) feeding behavior (i.e., with active prey selection) or to opportunistic feeding on available prey is still open to discussion (many references report filter feeding as complementary feeding behavior on this species). Accordingly, I recommend changing the statement to something like '...anchovy opportunistically feed on select species...' (the authors can use the same citation as in the previous version, and even include some more if they find it appropriate).

- *It is important to make the distinction between larval anchovy and adults as larval anchovy are not yet capable of filter feeding. However, we still agree with the reviewer's suggestion and have changed the sentence accordingly.*

L.207-209: According also to my previous comment, please change to '...is not surprising given the importance of Calanus in the diet of other larval fishese.g., 10, due to their high lipid and...'

- *Corrected.*

L.220: Please add reference(s) after '...is approximately four years'.

- *References to support the entire statement, not just maximal age, are listed at the end of the sentence.*

L.273: Please check the figure number, did the authors want to refer it as Figure 5?

- *Corrected.*

L.285: Please add some specific examples when mentioning 'predation'. Something like '....provisioning and predation (e.g., intraguild predatione.g., 55; [doi: 10.1007/s00227-011-1699-2], predator-prey interactionse.g., 58; [doi: 10.3354/meps208229], etc.) are likely co-contributors to... ' might be appropriate here. Please note I include four references, two already used, and two additional ones with doi, as those could summarize different scenarios as examples. Please feel free to include more if appropriate.

- *We added in predator-prey interactions. Further detail is already provided in the "Bottom-up or top-down effects" section. We appreciate the references. Unfortunately, we are at the maximum allowed number of citations but have included the two references already used.*

L.289: Please remove 'typically' from the sentence (this term was already used in previous line).

- *Corrected.*

L.296-300: According to my previous review, the authors still need to include more text regarding their methodological approach. I understand that a specific section might be too much, but at least some extended text in the Discussion regarding the major weaknesses of the methodology applied, limitations of the approach, and/or further potential improvements is required. Accordingly, in this part of the text the following issues must be treated (including appropriate references):

zooplankton sampling bias (due to sampling tools' limitations to get the whole size range fractions of the zooplankton community that might be potential prey for anchovy larvae);

- *We have now added additional text to the second paragraph of the "Bottom-up or top-down effects" section that underlines that smaller nauplii are not sampled with the 505um Bongo net and that early copepodite stages are not identified but that they are part of the diet. But the data does adequately resolve the population dynamics of the larger mature copepods also preyed on by the larvae.*

potential application of a multi-proxy approach to test the hypothesis (i.e., combining visual prey identification from stomach contents of larvae, SIA and DNA-metabarcoding; use citation doi: 10.1038/s41598-020-74602-y); the need of dedicated sampling designs (how would the test of the hypothesis be improved by a new sampling design? Which aspects should be improved?); and how routinary sampling programs might contribute to improve the available data to apply similar approach to other regions and/or species?

- *We have rephrased the last few sentences to: “ To advance mechanistic understanding of TEEL and the forecasting benefits such ecological indicators may provide, we need to combine visual and molecular approaches to better identify the main prey species, and further explore how and where FCL is regulated in the food chain, its relationship to upwelling dynamics and its direct impacts on vital rates and survival. To achieve this, we need to collect samples that better resolves gradients in larval and plankton communities and interactions across space or time. Future research should also be aimed at investigating the applicability of TEEL to other fish stocks and how FCL measurements may be implemented for species of interest in near real time.”*